# Mapping the dependence of BC radiative forcing on emission region and season

Petri Räisänen[1], Joonas Merikanto[1], Risto Makkonen[1,2], Mikko Savolahti[3], Alf Kirkevåg[4], Maria Sand[5], Øyvind Seland[4], and Antti-Ilari Partanen[1]

[1]Finnish Meteorological Institute, Climate Research, Helsinki, Finland
[2]University of Helsinki, Institute for Atmospheric and Earth System Research, Helsinki, Finland
[3]Finnish Environment Institute, Climate and Air Pollution, Helsinki, Finland
[4]Norwegian Meteorological Institute, Oslo, Norway
[5]CICERO Center for International Climate Research, Oslo, Norway

**Correspondence:** Petri Räisänen (petri.raisanen@fmi.fi)

**Abstract.**

For short-lived climate forcers such as black carbon (BC), the atmospheric concentrations, radiative forcing (RF) and, ultimately, the subsequent effects on climate, depend on the location and timing of the emissions. Here, we employ the NorESM1-Happi version of Norwegian Earth System Model to systematically study how the RF associated with BC emissions depends on the latitude, longitude and seasonality of the emissions. The model aerosol scheme is run in an offline mode, to allow for an essentially noise-free evaluation of the RF associated with even minor changes in emissions. A total of 960 experiments were run to evaluate the BC direct RF (dirRF) and the RF associated with BC in snow/ice (snowRF) for emissions in 192 latitude-longitude boxes covering the globe, both for seasonally uniform emissions and for emissions in each of the four seasons separately. We also calculate a rough estimate of the global temperature response to regional emissions, and provide a fortran-based tool to facilitate the further use of our results.

Overall, the results demonstrate that the BC RFs strongly depend on the latitude, longitude and season of the emissions. In particular, the global-mean dirRF normalized by emissions (direct specific forcing; dirSF) depends much more strongly on the emission location than suggested by previous studies that have considered emissions from continental/subcontinental-scale regions. Even for seasonally uniform emissions, dirSF varies by more than a factor of ten depending on emission location. These variations correlate strongly with BC lifetime, which varies from less than 2 days to 11 days. BC dirSF is largest for emissions in tropical convective regions and in subtropical and midlatitude continents in summer, both due to the abundant solar radiation and strong convective transport, which increases BC lifetime and the amount of BC above clouds. The dirSF is also relatively large for emissions in high-albedo high-latitude regions such as Antarctica and Greenland. The dependence of snow specific forcing (snowSF) on the emission location is even larger. While BC emissions originating from most low-latitude regions result in negligible snowSF, the maxima of snowSF for emissions in polar regions greatly exceed the largest values of dirSF for low-latitude emissions. The large magnitude of snowSF for high-latitude BC emissions suggests that, for a given mass of BC emitted, also the climate impacts are largest for high-latitude emissions.

The additivity of the RFs resulting from BC emissions in different regions and seasons is also investigated. It is found that dirRF is almost additive for current-day emissions: summing the RFs computed for individual regions/seasons without considering BC emissions from elsewhere overestimates dirRF by less than 10%. For snowRF, the overestimate is somewhat larger, ∼20%.

## 1   Introduction

Reductions in black carbon (BC) emissions are considered a viable option to mitigate climate change globally, but particularly over the Arctic (AMAP, 2015). A policy initiative "Enhanced Black Carbon and Methane Emission Reductions, and Arctic Council Framework for Action" (Arctic Council, 2015) is on track to limit Arctic BC emissions by 25–33% relative to 2013 levels by 2025 (Arctic Council, 2021). However, considerable uncertainties on the basic science regarding the efficacy of such measures remain. One of the key issues is that for short-lived climate forcers (SLCFs) such as BC, the atmospheric concentrations, the radiative forcing (RF), and ultimately, the effects on climate, depend on the location and timing of the emissions (Persad and Caldeira, 2018; Westervelt et al., 2020). Consequently, the development of robust emission metrics to guide climate policy is more complicated for SLCFs than for long-lived greenhouse gases like $CO_2$ (e.g. Collins et al., 2013; Kupiainen et al., 2019). This is particularly the case for BC, which influences the climate through a multitude of effects, including its direct radiative effects due to absorption and scattering of solar radiation (aka. aerosol-radiation interaction), changes in snow and sea-ice albedo, and semi-direct and indirect effects on clouds (Koch and Del Genio, 2010; Bond et al., 2013; Aamaas et al., 2017).

In this paper, we revisit the question of how the RF of BC depends on the location and season of the emissions, focusing on the direct radiative forcing (dirRF) of atmospheric BC and the RF due to BC in snow and ice (snowRF). This question has been previously addressed in several climate model and chemical transport model studies, for various continental/subcontinental-scale regions (Reddy and Boucher, 2007; Rypdal et al., 2009; Bond et al., 2011; Yu et al., 2013; Bellouin et al., 2016). Different metrics such as the Global Warming Potential (GWP; the ratio of the cumulative radiative forcing due to the instantaneous release of a given mass of pollutant over a time horizon to the same quantity for the same mass of emitted $CO_2$) or the Specific Forcing Pulse (SFP) (the energy added to the Earth-atmosphere system by one gram of a species X emitted in a region S during its entire lifetime, in units of $J\,g^{-1}$) have been used in these studies. These differences are however not essential in the present context, inasmuch as all the metrics normalize the RF by the mass of BC emitted per time. These studies have shown that the normalized BC dirRF can depend substantially on the emission region. Reddy and Boucher (2007) found GWP values ranging by a factor of ca. 1.8 depending on the emission region (largest for Africa and smallest for Europe) and Rypdal et al. (2009) reported almost as wide a range (largest for Africa and smallest for Russia). In an apparent contrast to these studies, the multimodel study of Yu et al. (2013) found a larger normalized BC dirRF for emissions from Europe than for emissions from North America, South Asia and East Easia, although it should be noted that their European emission region also contained parts of North Africa and the Middle East. Furthermore, a substantial intermodel variability was demonstrated in the normalized BC dirRF, with variations of ∼ ±50–70% depending on the emission region.

Rypdal et al. (2009) found that while the GWP associated with BC snowRF was smaller than that associated with dirRF, it varied by more than an order of magnitude between the emission regions. Bond et al. (2011) reported that the sum of SFPs for dirRF and snowRF due to energy-related BC emissions from different regions varied by ca. 45% but this variability would be larger if it was not for the compensation between the direct and cryospheric effects: while BC emitted from higher-latitude regions generally has a larger cryospheric forcing, it often has lower atmospheric forcing than lower-latitude emissions because of less deep convection and shorter atmospheric lifetime. Bond et al. (2011) also noted the impact of emission season. The SFP is enhanced for emissions in summer because deep convection lofts the BC above reflective clouds and because the aerosol absorbs more sunlight during the longer summer days.

Bellouin et al. (2016) compared, as a part of the multimodel ECLIPSE study, the impact of reducing BC emissions from Europe and East Asia. They found, for the sum of dirRF and the indirect RF (indirRF) due to aerosol-cloud interaction, that the RF normalized by emitted mass was larger for BC emission perturbations in summer than in winter, and slightly larger for European than East Asian emissions. They also reported (although based on a single model only) a negative RF contribution due to rapid adjustments in summer and a substantial positive snowRF especially for European BC emissions in winter, both of these factors counteracting the seasonal variation associated with the sum of dirRF and indirRF.

A common limitation of the aforementioned studies is the coarse resolution of the normalized RF estimates with respect to the size of the emission regions. Typically, a dozen or less continental/subcontinental regions were considered. In a radically different approach, Henze et al. (2012) applied an adjoint of the GEOS-Chem chemical transport model to evaluate the dirRF efficiency (global-mean dirRF associated with emissions in a given grid cell divided by the emission rate for that grid cell) of aerosols as a function of emission location at the model grid-point resolution ($2° \times 2.5°$). For BC, large dirRF efficiencies were found for emissions in polar regions and in the northern parts of North America and Eurasia. The results suggested that variations in surface albedo, and secondarily variations in BC loss rate, were the most important factors for the distribution of the BC dirRF efficiency. Overall, while the study by Henze et al. (2012) was based on relatively short simulations (12 one-week adjoint model runs) and only considered dirRF, it underscored the importance of a spatially refined treatment of the RF associated with BC emissions.

In the present work, a large set of experiments is conducted with the NorESM1-Happi climate model (Graff et al., 2019) to systematically evaluate the impact of emission location and season on the RF resulting from BC emissions. Similarly to Reddy and Boucher (2007); Rypdal et al. (2009); Bond et al. (2011) and Yu et al. (2013), we employ ordinary "forward" model simulations, but we consider a much higher spatial resolution for the emissions (a total of 192 emission regions). By using the NorESM1-Happi aerosol scheme (Kirkevåg et al., 2013; Graff et al., 2019) in an offline mode, an essentially noise-free evaluation of the RF is achieved. This allows a detailed view of the impact that emission latitude, longitude and season have on the associated RF for BC emissions, considering both dirRF and snowRF.

A basic assumption of our analysis approach is that the RFs associated with BC emissions in different regions and seasons are additive, which is strictly true only if the response of BC RF to changing emissions is perfectly linear. Therefore, to evaluate the reasonableness of this assumption, we also investigate the additivity of BC RFs resulting from emissions in different regions/seasons. One finding from this investigation is that for the BC indirRF simulated by NorESM1-Happi, additivity is

violated severely. For this reason, indirRF is left out from the main analysis addressing the impact of emission location and season. Nevertheless, it should be kept in mind that, while BC indirRF is subject to large uncertainties, it can signficantly alter the overall RF of BC (Kühn et al., 2020). In fact, for NorESM1-Happi, it amounts to roughly 25% of dirRF.

The structure of this paper is as follows. The model set-up and RF calculations are described in Sect. 2 and the experiments are introduced in Sect. 3. Next, to set the background for further analysis, Sect. 4 first briefly discusses the BC RF simulated by NorESM1-Happi in the case of a realistic global distribution of BC emissions. The additivity analysis on BC RF is also included here. The impact of emission location and season on the resulting BC dirRF and snowRF is then evaluated systematically in Sect. 5. The implications of the present work are discussed further in three subsections. First, the results are put in the context of previous studies in Sect. 6.1. Second, an (admittedly very rough) estimate for the climate effects arising from the BC RF associated with emissions in different regions is presented in Sect. 6.2. Third, some further analysis regarding the nonlinearity of BC RF is provided in Sect. 6.3. Conclusions follow in Sect. 7.

## 2 Methods

### 2.1 NorESM1-Happi and the model set-up

The experiments were conducted with the NorESM1-Happi model (Graff et al., 2019), which is an upgraded version of the NorESM1-M model used for CMIP5 (Bentsen et al., 2013; Iversen et al., 2013; Kirkevåg et al., 2013). In the current experiments, sea surface temperatures and sea ice fraction were prescribed based on observations (Hurrell et al., 2008) for years 2012–2017 and sea ice thickness was set to $2\,\mathrm{m}$ ($1\,\mathrm{m}$) for the northern (southern) hemisphere. The following model components were included:

- The atmospheric component is the Oslo version of the Community Atmosphere Model (CAM4-Oslo), which differs from the original CAM4 (Neale et al., 2010, 2013) through modified chemistry-aerosol-cloud-radiation interaction schemes (Kirkevåg et al., 2013). The finite-volume dynamic core is employed. The horizontal resolution is $1.9°$ in latitude and $2.5°$ in longitude, with 26 levels in the vertical and the model top at 2.19 hPa. An error in the aerosol life-cycle scheme was corrected for NorESM1-Happi (Graff et al., 2019). This acts to reduce the atmospheric residence time of BC and upper tropospheric BC concentration.

- The land component is the original version 4 of the Community Land Model (CLM4) of CCSM4 (Oleson et al., 2010; Lawrence et al., 2011), which includes the SNow, ICe and Aerosol Radiative model (SNICAR; Flanner and Zender, 2005, 2006). SNICAR simulates prognostically the snow grain size and the concentrations of mineral dust and hydrophobic and hydrophilic BC in snow, and uses this information to compute snow single-scattering properties, the snow albedo and absorption of solar radiation in snow.

- The sea ice model is CICE4 (Gent et al., 2011; Holland et al., 2012). While ice area fraction and ice thickness are prescribed in this work, the amount of snow on ice and the concentrations of absorbing aerosols (BC and dust) in snow and ice are computed interactively, and they influence the surface albedo over sea ice (Briegleb and Light, 2007).

– The model components are coupled through the CCSM4 coupler CPL7 (Craig et al., 2012).

In the present experiments, the aerosol life-cycle scheme in NorESM1-Happi is run in an offline mode (Kirkevåg, 2013).
The physical and chemical processes influencing aerosol concentrations are simulated interactively, but the simulated aerosol concentrations do not influence the simulated climate. Instead, in the atmosphere model, CCSM4's prescribed standard aerosols are used for integrating the model forward in time. In this study, we extended the offline treatment of aerosols to BC in snow. The BC concentration in snow is simulated interactively, based on BC deposition fields provided by NorESM's aerosol scheme. Snow albedo is computed both with and without BC; the latter is used for integrating the model forward in time, and the former for diagnostics. The offline approach is crucial for the present study, since it allows for an essentially noise-free evaluation of RFs associated with even minor changes in emissions. If, instead, the emissions were allowed to influence the simulation, the RFs could easily be swamped by radiative flux changes related to internal climate variability.

## 2.2 Radiative forcing calculations

A limitation of the offline approach is that it can only be used for evaluating BC instantaneous RF. Since BC does not influence the simulated temperature, the effect of rapid adjustments (including but not limited to the semi-direct effect of BC on clouds (Koch and Del Genio, 2010)) cannot be evaluated. The RFs are evaluated based on extra diagnostic calls to the model shortwave radiation scheme:[1]

$$\text{dirRF} = F_{\text{air+snow}} - F_{\text{snow}} \tag{1}$$

$$\text{snowRF} = (F_{\text{air+snow}} - F_{\text{air}}) - (F_{\text{air+snow,0}} - F_{\text{air,0}}) \tag{2}$$

$$\text{indirRF} = F_{\text{aie}} - F_{\text{aie,0}} \tag{3}$$

Here, $F$ denotes net (down−up) solar radiative fluxes, and the subindices "air+snow", "air" and "snow" refer to calculations in which BC is included in both air and snow, only in air, and only in snow, respectively, and "aie" refers to a calculation targeted at evaluating the aerosol indirect effect on liquid-phase clouds. Note that for snowRF, the additional pair of terms on the rhs ($F_{\text{air+snow,0}} - F_{\text{air,0}}$) is the BC snowRF *in the absence of BC emissions*. It is nonzero ($\approx 0.004$ W m$^{-2}$ in the global mean) because in the model initial conditions, there is some BC stored in snow in regions with permanent snow, mainly in Greenland. Likewise, $F_{\text{aie}}$ for the experiment with no BC emissions has to be subtracted to define BC indirRF. Further details on how $F_{\text{air+snow}}$, $F_{\text{air}}$, $F_{\text{snow}}$, and $F_{\text{aie}}$ are calculated are provided in Appendix A.

While Eqs. (1–3) can be applied to define the RFs at any level in the atmosphere at any spatiotemporal scale, we will only consider here annual-mean RFs at the top of the atmosphere (TOA). Furthermore, we found that indirRF depends very nonlinearly on BC emissions and pre-existing aerosol concentrations, which would make the interpretation of the experiments discussed in Sect. 3 difficult. Therefore, the focus of this study will be on BC dirRF and snowRF.

---

[1]The model also includes a calculation of longwave indirRF for liquid-water clouds. This effect is however very small (nearly two orders of magnitude smaller than the shortwave indirRF for BC), and for simplicity it has been ignored throughout this paper.

## 2.3 Treatment of aerosol emissions other than BC

The chemical components in the NorESM1 aerosol scheme (Kirkevåg et al., 2013) include BC, sulphate, organic matter (OM), sea salt, and mineral dust. The BC emissions are discussed in Sect. 3 below. For the other aerosol components, a realistic spatiotemporal distribution of emissions was aimed at. The emissions follow Kirkevåg et al. (2013), with modifications for sulphur dioxide $SO_2$ (a precursor of sulphate) and OM. The $SO_2$, OM as well as BC emission sources are divided in NorESM1 into "fossil fuel" and "biomass burning" emissions, with somewhat different treatment of mixing and other physicochemical processes in the aerosol scheme (see Kirkevåg et al., 2013, Fig. 1). For the biomass burning $SO_2$ and OM emissions, RCP8.5 emissions (Lamarque et al., 2011; Riahi et al., 2011) for year 2015 are assumed in this study. Correspondingly, the fossil fuel $SO_2$ and OM emissions are taken from the ECLIPSE V6b Current Legislation (CLE) data (IIASA (2019); see also Klimont et al. (2017)) for year 2015, with the agricultural waste burning (AWB) sector eliminated due to overlap with biomass burning emissions. Sensitivity tests indicated that the impacts of $SO_2$ and OM emissions on the BC burden simulated by NorESM1-Happi are very small. Likewise, BC emissions have very little impact on the burdens of $SO_4$ and OM.

## 3 Experiments

**Table 1.** List of NorESM1-Happi experiments

| Experiment(s) | Description |
| --- | --- |
| REAL | A realistic distribution of BC emissions (global annual-mean value ca. $5.2 \cdot 10^{-13}\,\mathrm{kg\,m^{-2}\,s^{-1}}$, or $8.4\,\mathrm{Tg\,yr^{-1}}$) |
| IDEALIZED | An array of 960 experiments. BC emissions set to $10^{-12}\,\mathrm{kg\,m^{-2}\,s^{-1}}$ for one of 192 lat-lon boxes, |
| | either throughout the year or for one meteorological season; otherwise zero BC emissions |
| ZERO | Zero BC emissions. Needed for the computation of snowRF and indirRF (Eqs. 2 and 3). |
| COARSE | As REAL, but BC emissions modified so that they can be represented exactly by the reconstruction (Eq. 4) |
| REAL_X | 9 experiments with BC emissions in REAL multiplied by a constant factor X ranging from 0.1 to 10. |
| UNIF_Y | 11 experiments with a globally uniform BC emission rate Y ranging from $5 \cdot 10^{-14}$ to $5 \cdot 10^{-12}\,\mathrm{kg\,m^{-2}\,s^{-1}}$ |

The NorESM1-Happi experiments conducted in this work are listed in Table 1. The experiments differ only in terms of BC emissions; changes in emissions of co-emitted species are not considered. First, to set the background for further analysis, an experiment named REAL was conducted with a realistic distribution of BC emissions. ECLIPSE V6b CLE emissions (without the AWB sector) were used for fossil fuel BC and RCP8.5 emissions for biomass burning BC, with the emission year set to 2015. The global annual-mean BC emission rate is ca. $5.2 \cdot 10^{-13}\,\mathrm{kg\,m^{-2}\,s^{-1}}$. The fossil-fuel emissions were mostly released in the lowermost model layer, while biomass burning emissions, which comprise ca. 30% of the total, were distributed from the surface up to ca. 5 km height.

Second, the impact of spatial and seasonal distribution of emissions was addressed through a large array of experiments with an idealized distribution of BC emissions. Earth was split into 16 latitude zones and 12 longitude zones, which yields a total

of 192 latitude-longitude (briefly, lat-lon) boxes, each covering 72 model grid cells. The boundaries between the latitude zones (90.00°S, 79.58°S, 68.21°S, 56.84°S, 45.47°S, 34.11°S, 22.74°S, 11.37°S, 0.00°N, and correspondingly for the northern hemisphere) and those between the longitude zones (178.75°W, 148.75°W … 151.25°E, and 181.25°E) were chosen to match NorESM grid cell boundaries, so that each grid cell belongs to exactly one lat-lon box. The BC emissions either occurred uniformly over the annual cycle (ANN), or they were limited to one meteorological season (DJF, MAM, JJA, or SON). Together with the 192 options for the spatial distribution, this yielded 960 model experiments. In each experiment, the BC emission rate was set to an (arbitrary but typical) value of $\epsilon_{\mathrm{ideal}} = 10^{-12}\ \mathrm{kg\,m^{-2}\,s^{-1}}$ within one of the 192 lat-lon boxes for the emission season considered, and otherwise it was set to zero. BC was released to the lowest model layer, and it was treated as fossil-fuel BC. In addition, an experiment with zero BC emissions was conducted to evaluate BC snowRF and indirRF (Eqs. 2 and 3).

Third, the COARSE experiment was based on REAL, but the BC emissions were coarse-grained by averaging them over each of the 192 lat-lon boxes mentioned above and over each of the four meteorological seasons. All BC emissions (including those treated as biomass burning BC in REAL) were treated as fossil-fuel BC and released into the lowermost model layer, similar to the idealized experiments. The motivation for this experiment is explained in Sect. 4.2.

Fourth, for further analysis on the (non)linearity of BC RF wrt. the emission strength (discussed in Sect. 6.3), two smaller sets of experiments were conducted. In the 9 REAL_X experiments, the emission distribution employed in REAL was scaled by a globally uniform factor between 0.1 and 10. In the 11 UNIF_Y experiments, a globally uniform BC emission rate ranging from $5 \cdot 10^{-14}$ to $5 \cdot 10^{-12}\ \mathrm{kg\,m^{-2}\,s^{-1}}$ was assumed.

All experiments were run for six years (2012–2017) using sea surface temperatures and sea ice based on Hurrell et al. (2008), greenhouse gas concentrations following the RCP8.5 scenario, and aerosol emissions as described above. The last five years were included in the analysis.

## 4 Evaluation of methodology

### 4.1 BC radiative forcing in NorESM1-Happi for realistic emissions

We first briefly evaluate the experiment REAL in which a realistic global spatiotemporal distribution of emissions is assumed. Figures 1a and 2a show the annual-mean dirRF and snowRF at the TOA for REAL. The global-mean dirRF is $0.651\ \mathrm{W\,m^{-2}}$, which is close to the observationally constrained best-estimate of $0.61\ \mathrm{W\,m^{-2}}$ (90% confidence interval 0.16 to $1.40\ \mathrm{W\,m^{-2}}$) derived by Wang et al. (2016). The global-mean snowRF in REAL is $0.0285\ \mathrm{W\,m^{-2}}$. This is closer to the lower end of the range of instantaneous snowRF quoted in IPCC AR5 ($0.02$–$0.09\ \mathrm{W\,m^{-2}}$) (Myhre et al., 2013) while falling between the post-AR5 model-based estimates of $0.013\ \mathrm{W\,m^{-2}}$ by Lin et al. (2014) and $0.045\ \mathrm{W\,m^{-2}}$ by Namazi et al. (2015). [2] The global-mean BC burden in REAL is $2.78 \cdot 10^{-7}\ \mathrm{kg\,m^{-2}}$ (equivalent to 0.142 Tg) while the global-mean BC lifetime is 6.2 days. These values are slightly above the multi-model median (BC burden $0.131 \pm 0.047$ Tg; BC lifetime $5.5 \pm 1.9$ days; median$\pm$standard deviation) of the ensemble of 14 AeroCom phase III model simulations for year 2010 analyzed by Gliß et al. (2021).

---

[2]The IPCC AR5 and Lin et al. (2014) estimates include only anthropogenic BC, while natural BC is also included in our RF values. However, this does not bias the comparison substantially because in the REAL experiment, ca. 90% of snowRF is associated with fossil-fuel BC emissions.

Overall, the global-mean BC dirRF, snowRF, burden and lifetime in REAL are well within the range of previous studies. Even so, the results may be subject to model-specific biases. One known issue is that NorESM1-Happi likely overestimates upper-tropospheric BC concentrations, although less severely than NorESM1-M (Fig. S1 in Graff et al., 2019) (see also Samset et al., 2013; Allen and Landyut, 2014). This may be caused by overly efficient convective transport of BC to the upper troposphere (Kirkevåg et al., 2013; Allen and Landyut, 2014; Park and Allen, 2015; Sand et al., 2015). Since upper-tropospheric BC
contributes disproportionately to dirRF (Samset and Myhre, 2011; Samset et al., 2013) it appears likely that NorESM1-Happi somewhat overestimates dirRF, especially for BC emissions in convective regions.

## 4.2   Additivity and linearity of BC radiative forcings

In Sect. 5, the dependence of BC RF on emission region and season is evaluated based on idealized experiments, in which BC emissions are limited to a single region/season, with a fixed emission rate $\epsilon_{\text{ideal}} = 10^{-12}\,\text{kg}\,\text{m}^{-2}\,\text{s}^{-1}$. This approach involves
the assumptions that the BC RF is linear wrt. the emission rate and additive wrt. emissions from different regions/seasons. To evaluate to which extent these assumptions are violated in practice, we apply the results from the idealized experiments for reconstructing the BC RF in cases in which the emissions occur all over the globe and throughout the year. The reconstructed RFs ($\text{RF}_{\text{reconst}}$) are calculated from the idealized experiments as follows:

$$\text{RF}_{\text{reconst}} = \sum_{i=1}^{12}\sum_{j=1}^{16}\sum_{s=1}^{4}\frac{\epsilon_{\text{ijs}}}{\epsilon_{\text{ideal}}}\text{RF}_{\text{ideal},ijs}, \tag{4}$$

where $i$, $j$ and $s$ are indices for longitude, latitude and season, $\text{RF}_{\text{ideal},ijs}$ is the RF in the idealized experiment with the constant BC emission rate $\epsilon_{\text{ideal}}$ applied in the lat-lon box (i,j) for the season $s$, and $\epsilon_{ijs}$ is the average BC emission rate in the same lat-lon box and season for the experiment whose RF is being reconstructed. Note that besides the RFs, this formula can also be applied to other BC-related fields such as emission and deposition rates and column BC burden.

The patterns of dirRF and snowRF reconstructed for the REAL experiment are shown in Fig. 1b and 2b. They each correlate
strongly with the dirRF and snowRF simulated in REAL, with spatial correlation coefficients of 0.952 and 0.919, respectively. Nevertheless the reconstruction errors for dirRF (Fig. 1d,g) as well as for snowRF (Fig. 2d,g) feature both substantial small-scale variations and larger-scale biases. The relative global-mean bias is small for dirRF ($-2.5\%$) but rather large and positive for snowRF ($+44.9\%$). The reconstruction errors in Figs. 1d,g and 2d,g arise from two distinct sources, which we denote as the "emission error" and the "additivity error".

The emission error arises because the reconstruction cannot represent accurately the distribution of emissions in REAL. First, the emissions are represented at $1.9° \times 2.5°$ spatial resolution and monthly temporal resolution in REAL, while in the idealized experiments used for reconstruction, the spatial resolution is ca. $11.4° \times 30°$ and the temporal resolution is three-monthly. Second, the idealized experiments treat all BC as fossil-fuel BC (see Kirkevåg et al., 2013, Fig. 1) and place the emissions in the lowermost model layer, while for REAL, ca. 30% of the emissions are released as biomass burning BC, with a different
treatment in the aerosol scheme and larger emission heights. Since the reconstructed emissions for REAL are equal to both the actual and reconstructed emissions for COARSE, the emission error can be evaluated as the difference between COARSE and REAL, shown in Figs. 1e,h and 2e,h. It explains most of the locally large positive/negative reconstruction errors seen in

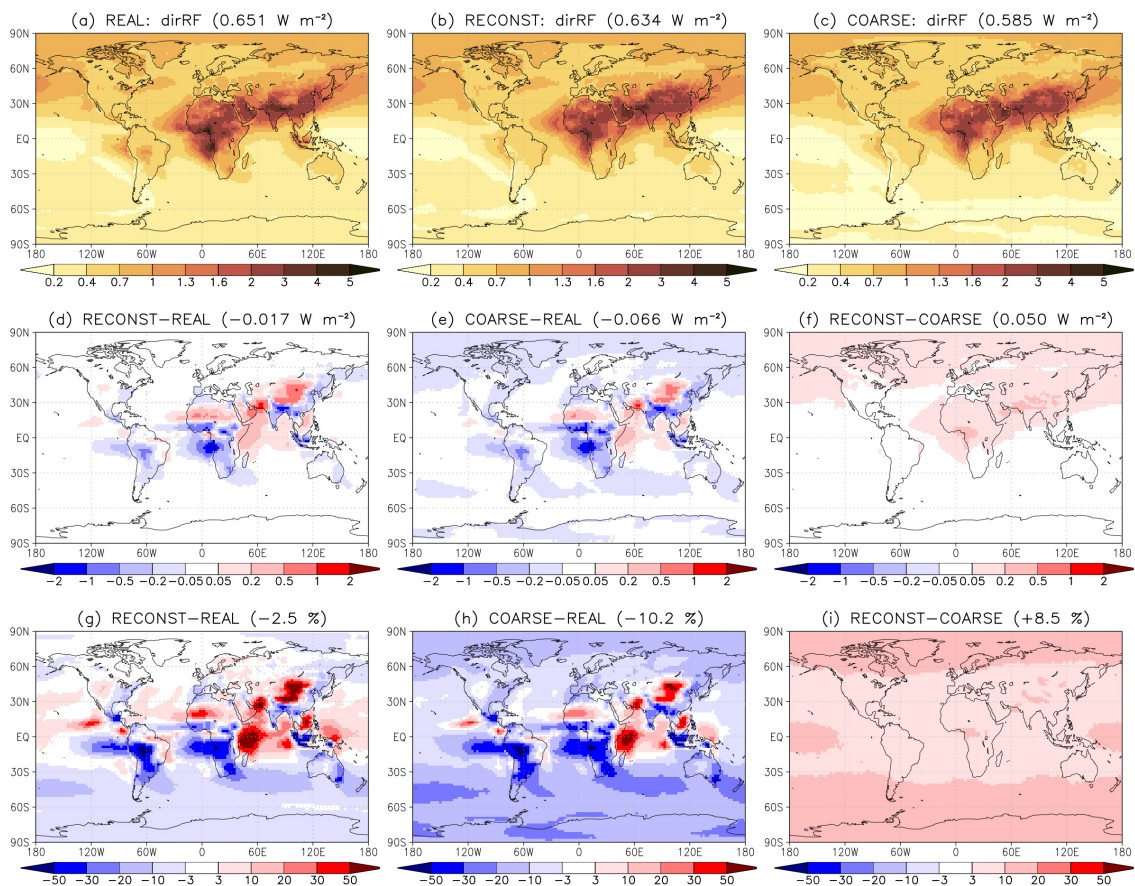

**Figure 1.** BC direct radiative forcing at TOA (W m$^{-2}$) in the experiments (a) REAL and (c) COARSE. Between them, (b) shows dirRF reconstructed using Eq. (4), which is identical for the two experiments. (d) Reconstruction errors (W m$^{-2}$) for REAL. These errors are further decomposed into two parts, where (e) the difference COARSE−REAL represents the emission error, arising from the inability of the reconstruction to represent accurately the emissions in REAL, and (f) the difference RECONST−COARSE (i.e., the reconstruction error for COARSE) is the additivity error that arises from nonadditivity and nonlinearity when combining the effect of BC emissions from different regions and seasons. (g)–(i) Relative errors corresponding to (d)–(f) (in %). Global-mean values are indicated in the panel titles.

Figs. 1d,g and 2d,g. It also influences the global mean values, especially for snowRF, for which it explains roughly one half of the reconstruction bias (0.0062 out of 0.0128 W m$^{-2}$). This bias mainly originates from the Tibetan region and is associated
with overestimated BC deposition (Figs. S2d,e,g,h and S3d,e,g,h in the Supplementary material) due to misrepresentation of the spatial emission patterns from India and China by the reconstruction (Fig. S1d,e,g,h). Furthermore, the emission error results in underestimated dirRF in regions where BC mostly originates from biomass burning in REAL, especially over Angola and off its western coast, and over the western Amazonas and downwind of it in the eastern Pacific (Fig. 1d,e,g,h).

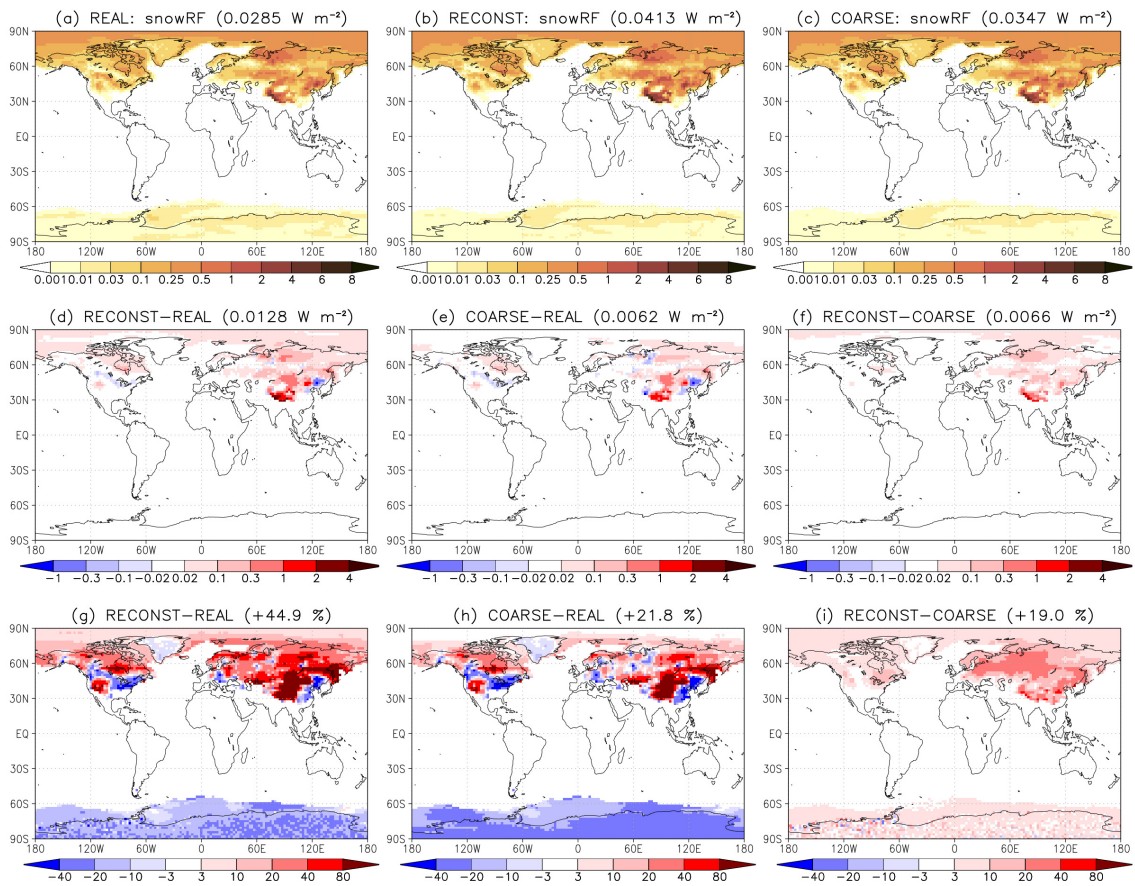

**Figure 2.** Same as Fig. 1 but for the TOA radiative forcing due to BC in snow.

The additivity error is more fundamental than the emission error in the sense that it addresses directly the validity of the basic
assumption of the reconstruction approach: that is, how large is the impact of nonadditivity and nonlinearity when combining
the effect of BC emissions from different regions and seasons? In other words, how large are the reconstruction errors if
the emission errors are eliminated? This question can be answered by considering the reconstruction errors for the COARSE
experiment, shown in Figs. 1f,i and 2f,i. Note that, by the design of the experiment, the reconstructed fields for COARSE are
identical to those for REAL (Figs. 1b and 2b) and are not shown separately.

For the COARSE experiment, the reconstruction reproduces the spatial patterns of dirRF and snowRF extremely well, with
correlation coefficients of 0.9996 and 0.996, respectively. The reconstruction errors are relatively smooth and positive almost
throughout (Figs. 1f,i and 2f,i). For dirRF, the global-mean reconstruction bias is 8.5%, in contrast to the slightly negative bias
of $-2.5\%$ for the reconstruction estimate of REAL. For snowRF, the global-mean reconstruction bias (19.0%) is significantly
smaller than that for the REAL experiment (44.9%). The positive errors suggest that dirRF and, in particular, snowRF, increase
slightly weaker than linearly with BC emissions. This is demonstrated explicitly in Fig. 8 below. We argue in Sect. 6.3 that this

occurs primarily due to partial saturation of BC absorption. For dirRF, a slight overestimate of atmospheric BC burden in the reconstruction (on average 2.6%; see Fig. S4i) also contributes to the positive reconstruction errors.

The above reconstruction tests, especially for the COARSE experiment, indicate that the separate consideration of emissions from different regions and seasons tends to result in overestimated RF, and more so for snowRF than dirRF. Nevertheless, spatial patterns of RF are reproduced very well, and the magnitude of the bias (on average below 10% for dirRF and below 20% for snowRF) appears reasonably small. It is concluded that the separate consideration of emissions from different regions/seasons in the idealized experiments provides acceptable results for dirRF and snowRF. In contrast, the BC indirRF diagnosed by NorESM1-Happi depends strongly nonlinearly on the emissions. The reconstruction overestimates the global-mean values severely, by 410% for REAL and by 228% for COARSE (Fig. S5). Accordingly, indirRF is omitted from the main analysis addressing the impact of emission location and season. However, some more discussion on the BC RF nonlinearity, including also indirRF, is provided in Sect. 6.3 below.

## 5   Results

This section has three subsections. First, in Sect. 5.1, we illustrate our approch by considering the RF associated with BC emissions from a single lat-lon region, roughly covering the Fennoscandian area. Next, in Sect. 5.2 the impact of emission location on BC RF is addressed systematically. Third, in Sect. 5.3, the impact of emission season is considered.

### 5.1   Example: Radiative forcing for emissions from Fennoscandia

Figure 3 displays the spatial distribution of BC burden, BC deposition rate, dirRF, and snowRF for the experiment in which a seasonally uniform BC emission rate of $10^{-12}\,\mathrm{kg\,m^{-2}\,s^{-1}}$ was assumed for the lat-lon box 56.84–68.21°N, 1.25–31.25°E. This area covers most of the Fennoscandian region and some of its surroundings. Not surprisingly, the simulated BC burden (Fig. 3a) peaks in the emission region, with a local maximum of ca. 88 $\mathrm{mg\,m^{-2}}$. Some BC is, however, transported further away (especially towards east/north-east due to the dominant wind directions), and the BC burden exceeds $1\mathrm{mg\,m^{-2}}$ (i.e., roughly 1% of the maximum value) in most of the Arctic, in parts of northern midlatitudes, and in northern Africa. The pattern of BC deposition is broadly similar to that of BC burden, although slightly more confined to the vicinity of the emission region (Fig. 3b). The dirRF follows largely the same pattern as BC burden, with some modulation due to, for example, differences in surface albedo, how much of BC is above clouds, and the availability of solar radiation (Fig. 3c; see also Fig. S6). The snowRF is constrained both by the distribution of BC deposition and the availability of snow (Fig. 3d).

Due to the offline treatment of aerosols, the simulated patterns of RF are quite smooth even though the RFs themselves are very small: the global-mean values are $1.90\,\mathrm{mW\,m^{-2}}$ for dirRF and $0.86\,\mathrm{mW\,m^{-2}}$ for snowRF. Of course, these RF values depend on the area $A$ of the lat-lon boxes used for emissions and on the emission rate $\epsilon$ employed. For this reason, we will consider global-mean specific forcings (SF), defined as the global annual-mean RFs ($\mathrm{RF_{glob}}$) normalized by the global

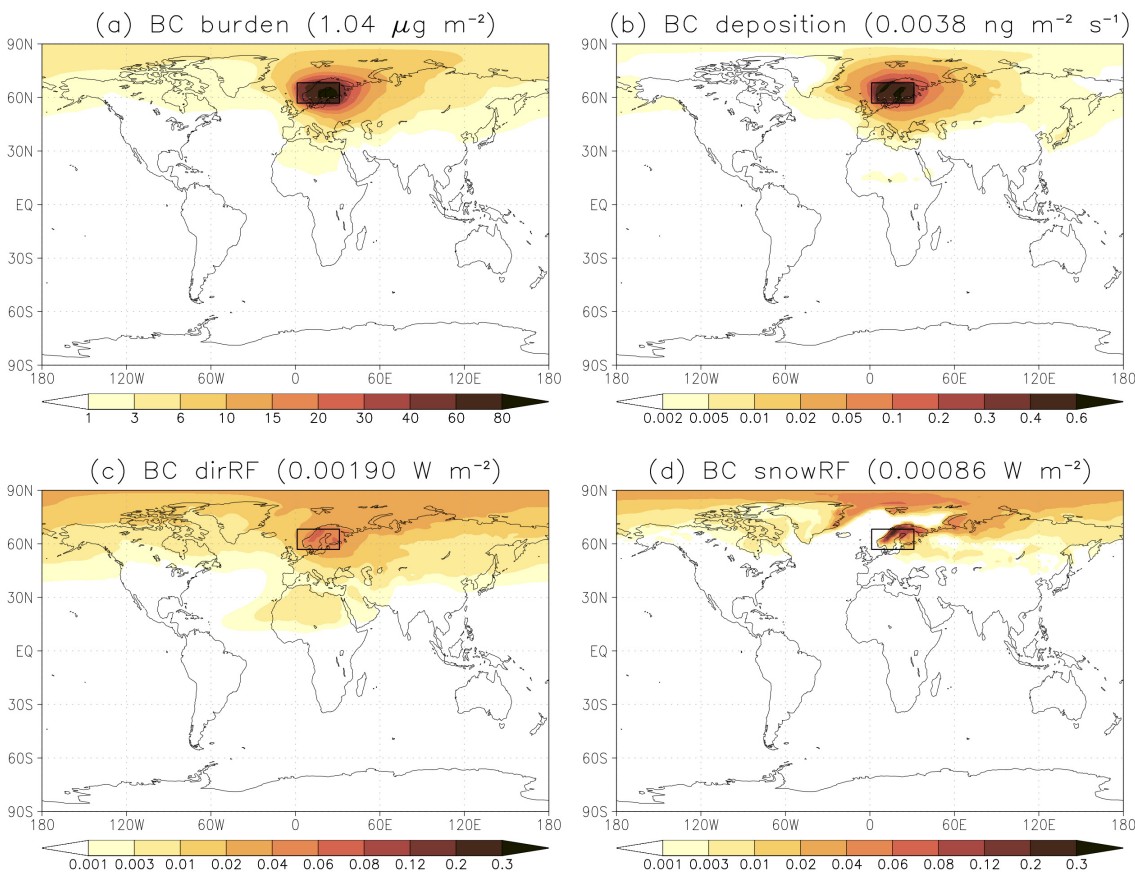

**Figure 3.** Annual-mean fields in the experiment in which a constant BC emission rate of $10^{-12}$ kg m$^{-2}$ s$^{-1}$ was applied in the lat-lon box 56.84–68.21°N, 1.25–31.25 °E (shown with a rectancle). (a) BC burden (in units of µg m$^{-2}$=$10^{-9}$ kg m$^{-2}$); (b) total BC deposition rate (in units of ng m$^{-2}$ s$^{-1}$=$10^{-12}$ kg m$^{-2}$ s$^{-1}$); (c) BC direct radiative forcing (W m$^{-2}$); and radiative forcing due to BC in snow (W m$^{-2}$). Global-mean values are given in the panel titles (in parentheses).

annual-mean emission rate $\epsilon_{\mathrm{glob}}$ resulting from emissions in a given region:

$$\mathrm{dirSF} = \mathrm{dirRF}_{\mathrm{glob}}/\epsilon_{\mathrm{glob}} \tag{5}$$

$$\mathrm{snowSF} = \mathrm{snowRF}_{\mathrm{glob}}/\epsilon_{\mathrm{glob}} \tag{6}$$

$$\epsilon_{\mathrm{glob}} = \epsilon f_{\mathrm{time}} A/A_{\mathrm{glob}}, \tag{7}$$

where $f_{\mathrm{time}}$ is the time fraction with emissions (1 for annual emissions and ca. 0.25 for seasonal emisions) and $A_{\mathrm{glob}}$ is the global surface area. This is similar to the concept of specific forcing pulse (SFP) introduced in Bond et al. (2011), but we leave out the word "pulse" because the specific forcings refer here to continuous emissions. The SFs will be expressed in units of TJ kg$^{-1}$ (i.e., the amount of energy added to the climate system in terajoules per 1 kg of BC emitted). BC burdens will likewise

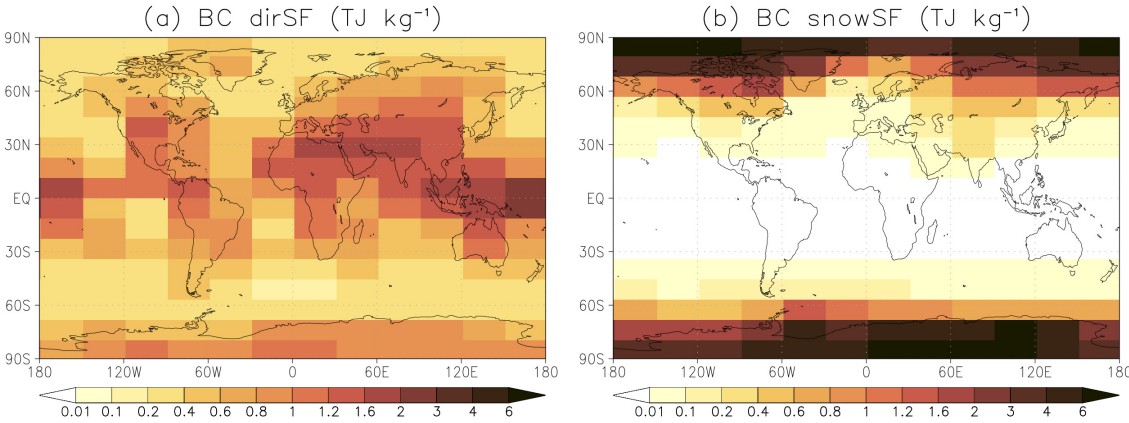

**Figure 4.** Global-mean values of (a) BC direct specific forcing and (b) specific forcing due to BC in snow $(\mathrm{TJ\,kg^{-1}})$ for the experiments in which uniform BC emissions over the year were assumed, separately for emissions in each of the 192 lat-lon boxes.

be normalized by the emissions, which yields BC lifetime, expressed in days. In the case of Fig. 3, the normalized global-mean

values are as follows: $0.50\,\mathrm{TJ\,kg^{-1}}$ for dirSF, $0.23\,\mathrm{TJ\,kg^{-1}}$ for snowSF, and 3.15 days for the BC lifetime.

### 5.2    Impact of emission location

Figure 4 summarizes the dependence of BC SF on the emission location, based on the experiments in which BC emissions are uniform over the year. Each lat-lon box is colored according to the global-mean SF associated with BC emissions within that lat-lon box; Fig. 4a showing BC dirSF and Fig. 4b, BC snowSF. (A corresponding figure for BC indirSF is provided in

the Supplementary material (Fig. S7) but it is not discussed here due to the strong nonlinearity noted above). The impact of emission location is very large, both for dirSF and snowSF. Considering first dirSF, the values vary by more than a factor of ten depending on the emission location, from $0.18\,\mathrm{TJ\,kg^{-1}}$ (for emissions at 45.47–56.84°S, 28.75°W–1.25°E) up to $2.33\,\mathrm{TJ\,kg^{-1}}$ (for emissions at 0–11.37°N, 151.25–181.25°E). The dirSF tends to be larger for emissions at low latitudes than at high latitudes; however, relatively large values also occur for emissions over Antarctica and Greenland. There are

also substantial variations as a function of longitude, dirSF being typically larger for emissions over land than over ocean. Nevertheless, the very largest values occur for emissions located east of Indonesia over the Tropical Warm Pool. This, together with the high values over low-latitude land areas, points to the important role of convective transport of BC.

The patterns of dirSF bear a strong resemblance to the corresponding patterns of BC lifetime, shown in Fig. 5a; the area-weighted spatial correlation between them is as high as 0.94. BC lifetime varies from less than 2 days for emissions in parts

of the Southern Ocean and the Arctic Ocean up to 11 days in the Tropical Warm Pool. Figure 5b further shows the ratio of dirSF to BC lifetime. This ratio varies by a factor of four, from 0.10 to $0.40\,\mathrm{TJ\,kg^{-1}\,day^{-1}}$, indicating that factors other than BC lifetime also play a substantial role for dirSF. BC dirSF is enhanced when BC is located above a high-albedo surface (or alternatively, above reflective clouds). The role of high surface albedo is seen in the locally large values of the dirSF-to-lifetime

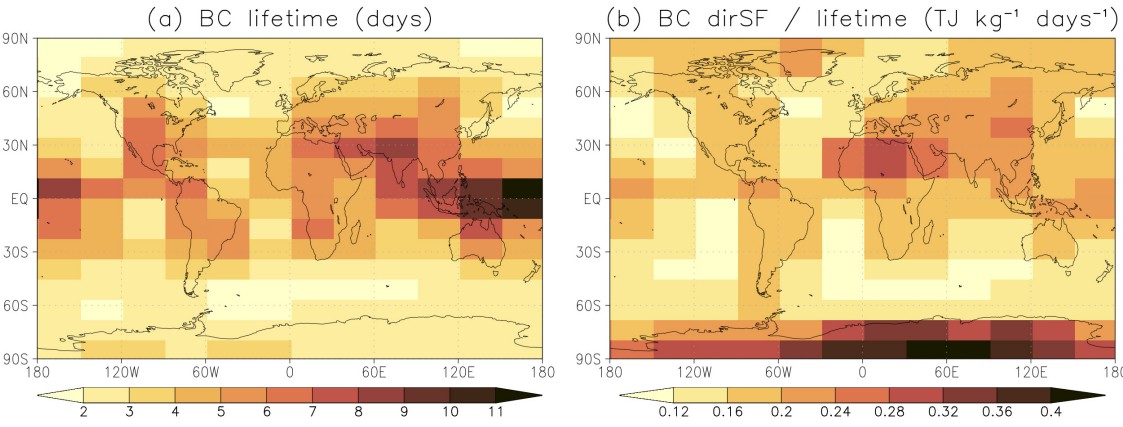

**Figure 5.** (a) BC lifetime (days) and (b) global-mean BC direct specific forcing divided by lifetime ($\mathrm{TJ\,kg^{-1}\,day^{-1}}$) for the experiments in which uniform BC emissions over the year were assumed, separately for emissions in 192 lat-lon boxes.

ratio for emissions over Greenland, Sahara, and especially Antarctica. Another factor that influences the dirSF-to-lifetime ratio
is the distribution of incoming solar radiation. Much of the emitted BC remains relatively close to the latitude of emission, and therefore, BC emitted at low latitudes is, on average, exposed to higher levels of solar radiation than BC emitted at high latitudes. This effect is, however, partially offset by the solar zenith angles being larger at high latitudes, which makes the slantwise aerosol optical depth larger for direct solar radiation.

For BC snowSF (Fig. 4b), the impact of emission location is even larger than that for dirSF. For low-latitude emissions
between $34.11°\mathrm{S}$ and $22.74°\mathrm{N}$, snowSF is minimal (generally below $0.01\,\mathrm{TJ\,kg^{-1}}$). Although some of the emitted BC is transported to the upper troposphere at high latitudes, its contribution to BC deposition on snow is negligible. The BC emissions from the northern India area ($22.74$–$34.11°\mathrm{N}$, $61.25$-$91.25°\mathrm{E}$) are associated with a slightly larger BC snowSF ($0.20\,\mathrm{TJ\,kg^{-1}}$), due to deposition in Tibet. In contrast, at high northern and southern latitudes, snowSF reaches very high values, with the maxima over the permanently snow/ice-covered regions of Antarctica and the Arctic Ocean exceeding $7\,\mathrm{TJ\,kg^{-1}}$. These maxima
greatly exceed the maximum of BC dirSF ($2.33\,\mathrm{TJ\,kg^{-1}}$, Fig. 4a). This suggests that for a given mass of BC emitted, BC total RF, and very likely, the climate effects, are largest for high-latitude emissions. This will be discussed further in Sect. 6.

It is also worth noting that BC snowSF shows a substantial dependence on the longitude of the emissions, for example, in the subarctic region ($56.84$–$68.21°\mathrm{N}$). There, snowSF varies from values close to $0.2\,\mathrm{TJ\,kg^{-1}}$ for emissions in the Atlantic and Fennoscandian regions to $1$–$1.2\ \mathrm{TJ\,kg^{-1}}$ for Siberian emissions and even up to $1.9\,\mathrm{TJ\,kg^{-1}}$ for emissions from northeastern
Canada. A key factor for these differences is how long snow persists into spring, since large snowSF requires a combination of extensive snow cover and abundant solar radiation in the regions where BC is deposited. A further factor is the shielding of snow by vegetation. The main BC deposition regions for Fennoscandian emissions (Fig. 3b) are characterized by both relatively early snowmelt and extensive forest cover, which both act to decrease the BC snowSF, as compared with Siberian and North American emissions originating from the same latitudes.

## 5.3 Impact of emission season

In this section, the impact of emission season on the global-mean BC dirSF, BC snowSF and BC lifetime is presented. Figure 6 summarizes the general features of the seasonal dependence. "Land-weighted" and "sea-weighted" zonal-mean values were defined as averages over the 12 emission regions in each latitude band weighted by the land fraction and sea fraction of the emission regions, respectively. The land-weighted and sea-weighted averages help to illustrate some differences between the impact of seasonal emissions over land and sea. This separation is however, not complete, because many of the 192 emission regions contain both land and sea areas. Detailed lat-lon maps showing the dependence on emission season are presented in the Supplementary material (Figs. S8–S10).

It is seen from Figs. 6a,b that BC dirSF depends substantially on the emission season. The dependence is particularly large at high and middle latitudes, owing to the strong seasonal cycle of insolation. BC dirSF is generally largest for emissions in the local summer and smallest for emissions in winter. As an exception, for emissions in the Arctic and in the southern parts of Southern ocean, dirSF peaks already for emissions in spring, presumably because snow and ice cover is larger in spring than in summer, and therefore, the surface albedo is higher. Furthermore, the dependence of dirSF on the emission season is generally larger for emissions over land than over sea. Over land, the impact of increased insolation is reinforced by increased BC lifetime, especially in the subtropics and at midlatitudes (Fig. 6e). This applies to emissions from (e.g.) the USA, southern/central Canada, northern India, China and southern/central Siberia in JJA (Fig. S10c), and emissions from Australia and southern Africa in DJF and SON (Fig. S10a,d). The increased lifetime is associated with lesser atmospheric stability and stronger convection in summer.

For BC snowSF (Fig. 6c,d), the impact of emission season is rather different from that for dirSF. The fundamental difference is that while the BC atmospheric lifetime is short (mostly one week or less), BC deposited on snow remains in the snowpack until snow melts. Therefore, BC deposited on snow in the dark season can absorb solar radiation in spring as long as there is snow left. This explains the relatively large annual-mean BC snowSF associated with Arctic BC emissions in DJF (and for the high Arctic, also SON). The largest snowSF by northern hemisphere high-latitude emissions is however caused by emissions in spring, as they maximize the concentration of BC in the uppermost snow in spring. The snowSF for northern hemisphere emissions is smallest for emissions in summer (in fact, near zero except for Arctic emissions) due to the lack of snow. In contrast, for emissions in Antarctica, where snow cover is permanent, snowSF is largest for emissions in summer (DJF), followed by spring (SON).

## 6 Discussion

### 6.1 Comparison with previous work

The NorESM1-Happi experiments indicate that the global-mean SF associated with BC emissions depends greatly both on the location and season of the emissions. The findings regarding emission season (larger dirSF for BC emissions in summer than in winter, and vice versa for snowSF except for permanently snow-covered regions) are largely consistent with previous

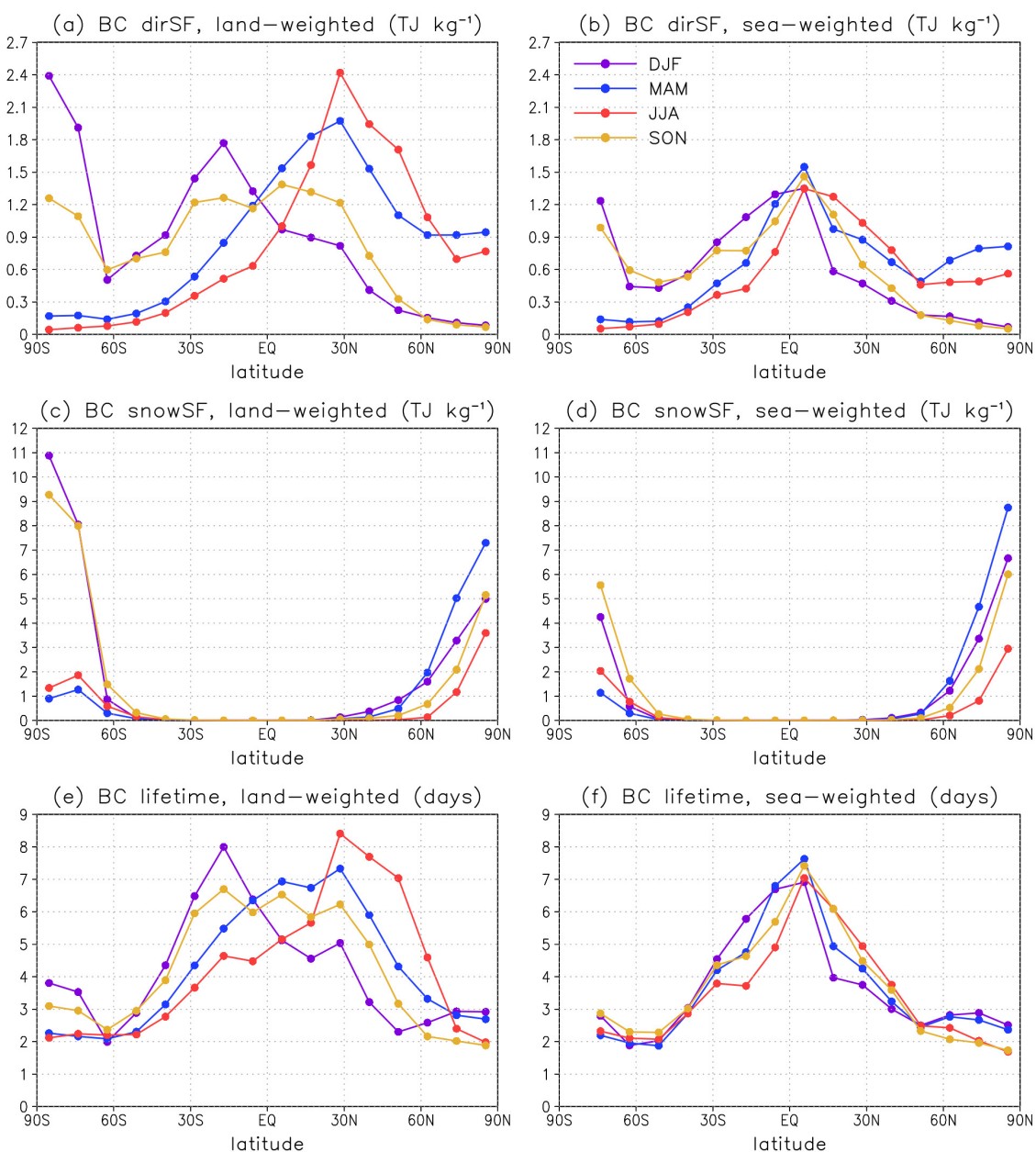

**Figure 6.** Global-mean BC direct specific forcing $(\mathrm{TJ\,kg^{-1}})$ associated with BC emissions in different seasons, zonally averaged over the 12 emission regions per latitude band and weighted by (a) land fraction and (b) sea fraction of the emission regions. (c) and (d): same as (a) and (b) but for the specific forcing due to BC in snow $(\mathrm{TJ\,kg^{-1}})$. (e) and (f): same as (a) and (b) but for BC lifetime (days).

work (Bond et al., 2011; Bellouin et al., 2016). However, even when considering seasonally uniform emissions, dirSF ranges

by more than a factor of ten depending on the emission location. This is a much larger range than found in previous studies, where the reported interregional differences have been less than a factor of two (Reddy and Boucher, 2007; Rypdal et al., 2009; Bond et al., 2011; Yu et al., 2013; Bellouin et al., 2016). The origin of this difference probably mostly lies in the size of the emission regions considered. For continental/subcontinental regions, compensation between locations at which BC emissions give rise to low or high normalized dirSF is likely to occur. The large differences in how the emission regions (and the distribution of emissions within them) are defined makes a quantitative comparison with the aforementioned studies challenging. However, qualitatively, our Fig. 4a is consistent the findings of Reddy and Boucher (2007) and Rypdal et al. (2009) that the normalized dirRF is relatively large for African emissions and relatively small for European and Russian emissions, and with the notion that enhanced dirRF is associated with BC emissions in convective regions (Bond et al., 2011). The details sometimes differ. For example, while Fig. 4a suggests that BC dirSF is larger for East Asian than European emissions, Fig. 6 in Bellouin et al. (2016) shows broadly similar specific RF for airborne BC for these two emission regions. This comparison is slightly ambiguous because Bellouin et al. (2016) did not report separately the contributions from aerosol-radiation and cloud-radiation interaction. However, the difference could also be related to the experimental setup. Bellouin et al. (2016) considered 20% reductions from near-present baseline emissions (which makes the baseline substantially more polluted for East Asia), while we added equally strong BC emissions in each region on top of a zero-BC baseline case. Furthermore, in the results of Bellouin et al. (2016), snowSF is more than twice as large for European emissions as for Asian emissions, while in our results (Fig. 4b) this difference is less obvious.

Henze et al. (2012) provide estimates of BC dirRF efficiency at an even higher resolution than the present study. To make their results quantitatively comparable to ours, the values in their Fig. 1a should be normalized by the grid-cell area in GEOS-Chem, but qualitatively, substantial differences are apparent. In particular, in contrast to our results (and also those of Reddy and Boucher (2007), Rypdal et al. (2009) and Bond et al. (2011)), the BC dirRF efficiencies reported by Henze et al. (2012) are generally small for low-latitude emissions, especially over oceans but also over continents, which suggests a negligible role for the convective transport of BC. The root cause of this discrepancy remains unclear.

For the total SF (defined here as the sum of dirSF and snowSF), the impact of emission location is even larger than for dirSF. The values shown in Fig. 7a vary by a factor of more than 30 (from 0.27 to 8.8 $\mathrm{TJ\,kg^{-1}}$). While most real-world BC emissions take place at low-to-mid latitudes, the total SF (i.e., RF normalized by emitted mass) is largest for high-latitude emissions. This occurs because for high-latitude emissions, a substantial fraction of the emitted BC is deposited on snow and sea ice. Indeed, as shown in Fig. 7b, the large total SF resulting from high-latitude BC emissions is strongly dominated by snowSF, while for tropical and subtropical BC emissions, the contribution from snowSF is very small.

The longitude of the emissions also matters, which should be considered in climate policy. For example, for the subarctic region, snowSF (Fig. 4b) and the total SF (Fig. 7a) are substantially larger for BC emissions in Siberia and northern North America than for equally large emissions in Fennoscandia, due to the longer snow season and lesser vegetation masking in the former regions. In this respect, it is worth noting the role of BC emissions from Russian oil industry in Western Siberia. Not only are these emissions the largest high-latitude BC emission source in the ECLIPSE V6b CLE dataset (Fig. S1a), but they are also located in a region where snow cover prevails relatively long in spring, which gives rise to a large RF per emitted mass of

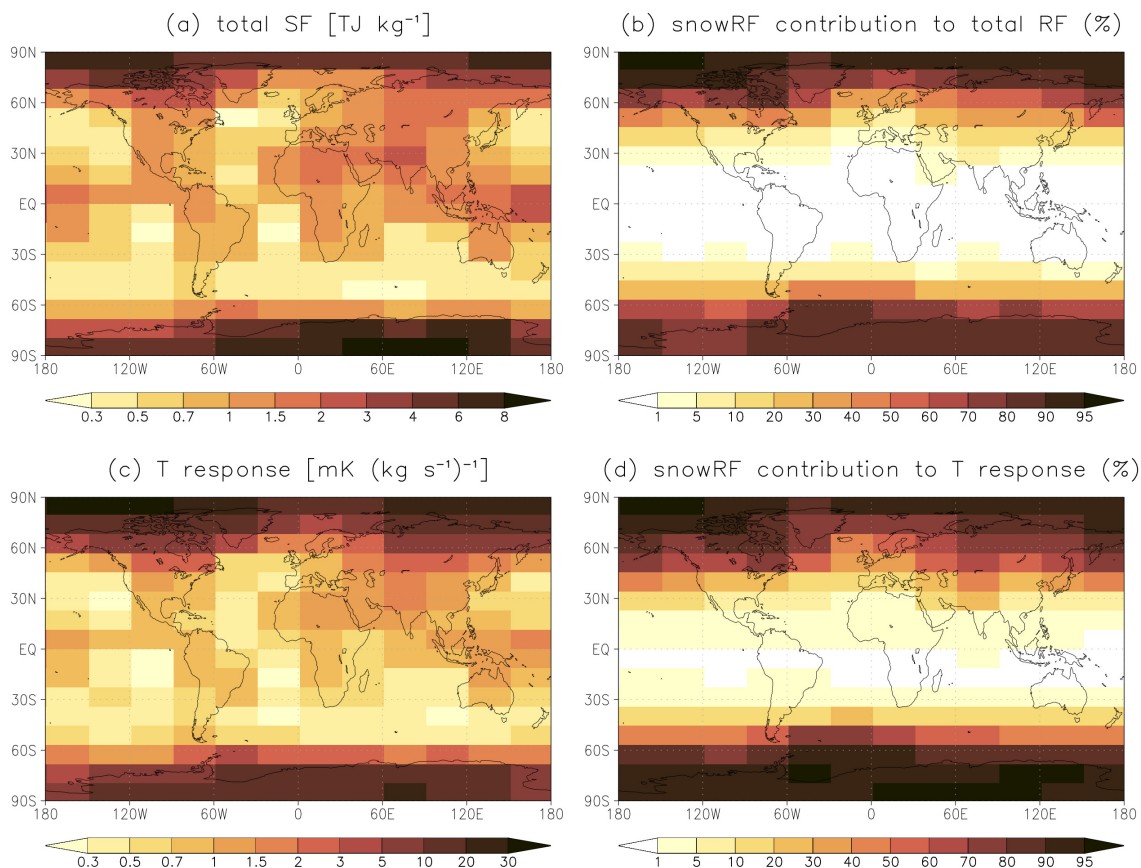

**Figure 7.** (a) Global-mean BC total SF, defined as the sum of dirSF and snowSF ($\mathrm{TJ\,kg^{-1}}$). (b) The fractional contribution of snowRF to the total RF (or equivalently, to the total SF) (%). (c) Estimated global-mean temperature response normalized by the BC emissions ($\mathrm{mK\,\left(kg\,s^{-1}\right)^{-1}}$). (d) The fractional contribution of snowRF to the temperature response (%). This figure is based on the experiments in which uniform BC emissions over the year were assumed, separately for emissions in 192 lat-lon boxes.

BC. This highlights the potential climate benefits of curbing the BC emissions from Russian oil industry (cf. Stohl et al., 2013). The role of Arctic BC emissions was also pointed out by Sand et al. (2016), who estimated the Arctic temperature response

to emissions of SLCFs from various regions and emission sectors. While they found that in absolute terms, the largest Arctic warming source stems from Asian BC emissions, the largest warming per unit emissions was associated with BC emissions from flaring and other activities within the Arctic nations.

## 6.2 Climate effects of BC emissions: a rough estimate

This study has focused on the instantaneous RF of BC only, including dirRF and snowRF. The problem with this is that for

BC, the instantaneous RF is, in fact, not a very good predictor of the ensuing climate response. Previous studies have shown

that the climate response to BC dirRF may depend strongly on the spatial distribution of BC, and especially its vertical profile (Hansen et al., 2005; Ban-Weiss et al., 2012; Flanner, 2013; Ocko et al., 2014; Samset and Myhre, 2015). In particular, Samset and Myhre (2015) demonstrated that while dirRF is largest when BC is located at high altitudes, the temperature response is largest for BC close to the surface. The disconnection between temperature response and dirRF is related to rapid adjustments in atmospheric stability, humidity and cloudiness. In general, for a realistic global distribution of BC, these adjustments act to make the BC effective radiative forcing (ERF) smaller than the instantaneous RF, thereby leading to a reduced global-mean temperature response (Hansen et al., 2005; Stjern et al., 2017; Smith et al., 2018; Richardson et al., 2019). This translates into low efficacy (the ratio of global-mean temperature responses to BC and $CO_2$ for a given RF) when defined wrt. the instantaneous or stratospherically adjusted RF (Hansen et al., 2005; Richardson et al., 2019). In contrast, high efficacy has been reported for BC snowRF (Hansen and Nazarenko, 2004; Hansen et al., 2005; Flanner et al., 2007) due to albedo feedbacks associated with accelerated snowmelt and due to warming concentrated near the surface.

Unfortunately, a direct evaluation of the climate response arising from BC emissions in the 192 lat-lon boxes and for the five seasonal distributions of emissions considered in this work would be unfeasible computationally. However, in order to get a rough estimate of the global-mean temperature response $\Delta T_{\mathrm{glob}}$, we resorted to the use of regional climate sensitivity coefficients (RCSs; aka. regional climate response coefficients) employed in previous research (Shindell and Faluvegi, 2009; Collins et al., 2013; Flanner, 2013; Sand et al., 2016). The RCSs describe the temperature response in four wide latitude bands to the dirRF and snowRF averaged separately over each of these bands (90–28°S, 28°S–28°N, 28–60°N, and 60–90°N), and they implicitly include the effect of rapid adjustments. The details of this analysis are presented in Appendix B.

Figure 7c shows $\Delta T_{\mathrm{glob}}$ normalized by BC emissions ($\Delta T_{\mathrm{glob,norm}}$; in units of $\mathrm{mK}\left(\mathrm{kg\,s}^{-1}\right)^{-1}$) for all 192 emission regions, and the fractional contribution to $\Delta T_{\mathrm{glob,norm}}$ by snowRF is displayed in Figure 7d. These results involve substantial uncertainties, not least due to the very coarse spatial resolution at which the RCSs were defined. Nevertheless, three qualitative conclusions appear robust. First, the estimated $\Delta T_{\mathrm{glob,norm}}$ depends very strongly on the emission location (Fig. 7c). Second, the largest $\Delta T_{\mathrm{glob,norm}}$ occurs for BC emissions in polar regions. Third, the large $\Delta T_{\mathrm{glob,norm}}$ resulting from high-latitude BC emissions is strongly dominated by BC snowRF (Fig. 7d) while for tropical and subtropical BC emissions, the contribution from snowRF is very small. Thus, the same general features apply to $\Delta T_{\mathrm{glob,norm}}$ as to the total SF and RF (Figs. 7a,b). In fact, $\Delta T_{\mathrm{glob,norm}}$ varies even more widely than the total RF, by a factor of over 160 (from 0.20 to 32.5 $\mathrm{mK}\left(\mathrm{kg\,s}^{-1}\right)^{-1}$). This mainly occurs because the impact of Arctic BC emissions is further exacerbated by the high efficacy of BC snowRF.

One uncertainty factor in our quantitative results is the assumption that the efficacy for BC snowRF is 3 times as large as that for BC dirRF (except for the Arctic temperature response to local BC RF in the Arctic, which was modelled explicitly by Flanner (2013)). Previous studies have indicated that the efficacy of BC snowRF defined wrt. the stratospheric adjusted RF (which is almost the same as the instantaneous RF) is well above 1; Hansen and Nazarenko (2004) give an efficacy of roughly 2, Hansen et al. (2005) 1.71, and Flanner et al. (2007) 2.1–4.5, depending on the experiment. Furthermore, as noted above, due to rapid adjustments, the efficacy of BC dirRF defined wrt. the stratospheric adjusted RF is very likely below 1. In particular, Hansen et al. (2005) report values of 0.58–0.78 for GISS ModelE-R, which Shindell and Faluvegi (2009) employed to derive the RCS coefficients used for BC dirRF here. Therefore the value of 3 for the snowRF-vs-dirRF efficacy ratio seems well

justified. Yet it might be an overestimate for regions with permanent snow (such as Antarctica and the interior of Greenland), where BC-induced snow cover changes are absent, and therefore, positive feedbacks due to albedo and lapse-rate changes are probably greatly reduced. If the snowRF-vs-dirRF efficacy ratio were set to 1, which is almost certainly too low, the normalized temperature response in Fig. 7c would be reduced by $\sim$30% for emissions in the Arctic and by 60-65% for emissions in the Antarctica; yet even then the qualitative features seen in Figs. 7c,d would remain largely the same.

Recently, Yang et al. (2019) and Sand et al. (2020) computed interactively the temperature response to regional BC emissions using CESM and NorESM1, respectively, although for much larger emission regions and strongly inflated emissions to improve the signal-to-noise ratio. When converted to the units of Fig. 7c, the values of $\Delta T_{\mathrm{glob,norm}}$ in Sand et al. (2020) range from 0.7–0.9 mK $\left(\mathrm{kg\,s^{-1}}\right)^{-1}$ for European emissions to ca. 0.5 mK $\left(\mathrm{kg\,s^{-1}}\right)^{-1}$ for South Asian emissions, while Yang et al. (2019) obtained values of 1.2 mK $\left(\mathrm{kg\,s^{-1}}\right)^{-1}$ for Arctic (60–90°N) emissions and 0.35 mK $\left(\mathrm{kg\,s^{-1}}\right)^{-1}$ for northern hemisphere midlatitude (28–60°N) emissions. While these studies show a tendency for higher-latitude BC emissions to cause a larger $\Delta T_{\mathrm{glob,norm}}$, broadly consistent with our results, especially the values obtained by Yang et al. (2019) are lower than those in Fig. 7c. This could be partially related to the use of much-larger-than-realistic BC emissions, which may lead to underestimated $\Delta T_{\mathrm{glob,norm}}$ (Sand et al., 2020) as well as underestimated SF (Fig. 8 below). However, much uncertainty also remains in the model-specific responses related especially to clouds.

The normalized temperature response in Fig. 7c can also be qualitatively compared with Aamaas et al. (2016, 2017), who computed Global Temperature Potential (GTP) and Absolute Regional Temperature Potential metrics for BC emissions in Europe and East Asia based on the RF data from Bellouin et al. (2016). Overall, their results suggest that the global-mean temperature is more sensitive to changes in European than East Asian BC emissions, especially if the enhanced efficacy of BC snowRF is taken into account (Aamaas et al., 2017). In contrast, our results rather show a slightly larger $\Delta T_{\mathrm{glob,norm}}$ for BC emissions in East Asia than in Europe. These differences are consistent with the respective RF differences to Bellouin et al. (2016) discussed above.

As a final check of our approach, we employed it for computing the BC contribution to the global-mean temperature (for details, see Appendix B). We obtained estimates of $\Delta T_{\mathrm{glob}}$=0.30 K for the year 2015 emissions (totalling 8.4 Tg yr$^{-1}$) employed in REAL and $\Delta T_{\mathrm{glob}}$=0.10 K for year 1850 emissions (totalling 3.1 Tg yr$^{-1}$) (Lamarque et al., 2011). This yields an estimate of 0.20 K for the BC-induced warming between 1850 and 2015. This value is within the uncertainty range given for the BC contribution to warming in the IPCC AR6 WG1 report (from ca.$-0.07$ K to 0.26 K for the temperature change from 1850–1900 to 2010–2019) (Fig. SPM.2 in IPCC, 2021) but closer to its upper bound.

## 6.3 Further discussion on the nonlinearity of BC radiative forcing

It was noted in Sect. 4.2 that summing the RFs computed for individual regions/seasons without considering BC emissions from elsewhere tends to slightly overestimate dirRF and especially snowRF. To provide some more insight on this finding, we consider in Fig. 8 experiments in which the spatiotemporal pattern of BC emissions was fixed but their magnitude varied. One set of experiments was based on REAL, but the emissions were scaled by a factor between 0.1 and 10, so that the global-mean emission rate varied between $5.2 \cdot 10^{-14}$ and $5.2 \cdot 10^{-12}$ kg m$^{-2}$ s$^{-1}$. In another set of experiments, uniform (UNIF) BC

emission rates ranging from $5.0 \cdot 10^{-14}$ kg m$^{-2}$ s$^{-1}$ to $5.0 \cdot 10^{-12}$ kg m$^{-2}$ s$^{-1}$ were considered. It is of note that for similar global-mean BC emissions, all quantities considered in Fig. 8 differ substantially between the REAL and UNIF experiments. This is consistent with the main result of our study that the BC RF depends strongly on the spatiotemporal distribution of the emissions. Here, we focus on the issue of nonlinearity.

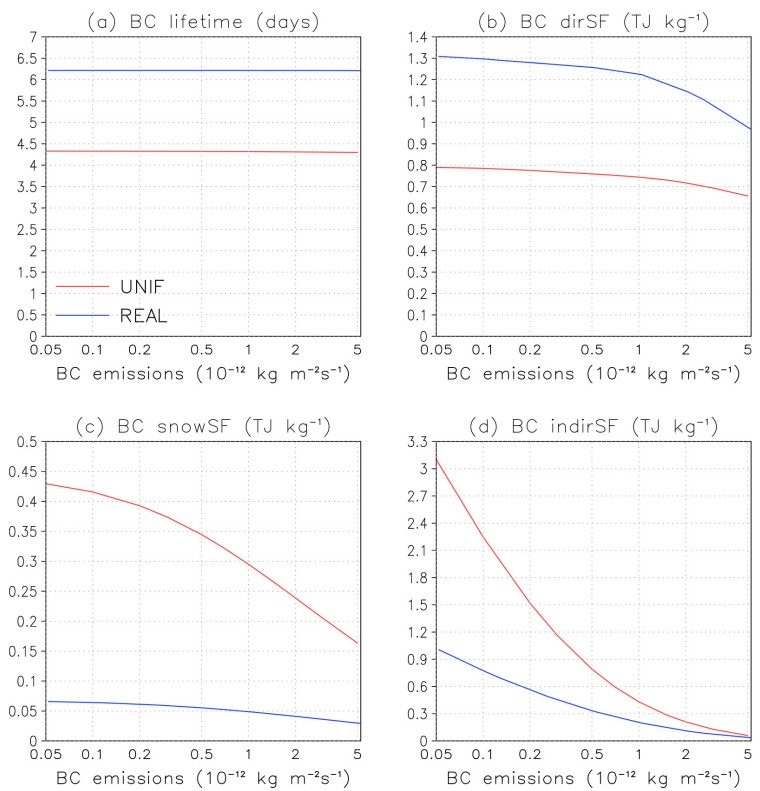

**Figure 8.** Global annual-mean values of (a) BC lifetime (days) and (b) BC direct specific forcing, (c) BC snow specific forcing and (d) BC indirect specific forcing (TJ kg$^{-1}$) for the experiments in which the spatiotemporal distribution of emissions was fixed but their magnitude was varied between roughly 0.1 and 10 times the present-day value. The experiments with uniform (UNIF) BC emissions are shown with red lines, and those with a realistic spatiotemporal emission distribution (REAL) with blue lines.

The global-mean BC lifetime is almost independent of the magnitude of emissions (Fig. 8a), which means that the BC
burden increases linearly with emissions. This result should be taken with caution due to the offline treatment of aerosols in our experiments. When BC is allowed to influence the simulated climate, the increase of BC burden with emissions is not necessarily linear (Sand et al., 2015, 2020; Yang et al., 2019).

BC dirSF shows a slight decrease with an increasing emission rate (Fig. 8b), and for snowSF, the decrease is stronger (Fig. 8c). For example, for emissions cut to one tenth of the realistic case REAL, dirSF is 4% larger and snowRF is 20%
larger, while for tenfold emissions, dirSF is smaller by 23% and snowSF is smaller by 47%. Since both the atmospheric BC

burden (Fig. 8a) and the BC burden in snow (not shown) increase quasi-linearly with emissions, this implies that both dirRF and especially snowRF increase weaker than linearly with increasing BC burden. Indeed, considering Beer's law (exponential decay of transmission with optical depth), a linear increase of absorption with increasing BC mass can only be expected at the limit of weak BC absorption, that is, when the BC absorption optical depth is small at all relevant wavelengths. With increasing BC burden, the absorption becomes partly saturated. This provides a plausible explanation for the tendency of the reconstruction to overestimate snowRF (Fig. 2f,i), and to a lesser extent, dirRF (Fig. 1f,i). These results are consistent with the snow albedo calculations by Flanner et al. (2007) and with the climate model simulations of dirRF by Chen et al. (2018).

Figure 8d confirms that the dependence of BC indirRF on the emission rate is strongly nonlinear. A number of factors could potentially contribute to this nonlinearity, including at least (1) the sublinearity of the Twomey effect (cloud optical depth increases weaker than linearly with increasing cloud droplet number concentration and cloud reflectance weaker than linearly with optical depth) (Twomey, 1977) and (2) the competition effect for water vapour (increasing availability of condensation nuclei acts to decrease the maximum supersaturation reached in the cloud) (Abdul-Razzak and Ghan, 2000; Kirkevåg et al., 2013). CAM5.1 simulations by Chen et al. (2018) (see their Table 1) also indicate some sublinearity of BC indirRF wrt. the magnitude of emissions, but not nearly as much as that seen for NorESM1-Happi.

Furthermore, for NorESM1-Happi, indirRF is positive (e.g., 0.168 W m$^{-2}$ for the REAL experiment), thereby reinforcing the effects of dirRF and snowRF. BC indirRF is known to strongly vary between different models, being negative in most but not all models (Bond et al., 2013; Cherian et al., 2017). It also depends heavily on the co-emitted aerosol precursors and aerosol species which can affect the aerosol hygroscopic properties and mixing state (Koch et al., 2011; Cherian et al., 2017). A possible explanation for the positive indirRF in NorESM1-Happi is that fossil-fuel BC can actually reduce the cloud droplet number concentration by acting as a condensation sink for sulfuric acid and water vapor while not being large enough to activate into cloud droplets (Koch et al., 2011; Blichner et al., 2021). In NorESM1-Happi, 90% of fossil-fuel BC mass is emitted to the nucleation mode with a diameter 23.6 nm and 10% to the fractal agglomerate mode with a diameter 200 nm, while in other models fossil-fuel BC is emitted at sizes between 30 nm and 100 nm (Mann et al., 2014). When numerous small fossil-fuel BC particles become coated with $SO_4$, they reduce the amount of condensable sulfuric acid available for the growth of aerosols into activation sizes, and they also become hygroscopic and act as a sink for water vapor. Thus, they can both inhibit the formation of large activation-size aerosol particles and reduce slightly the maximum supersaturation, which then further reduces the number of aerosol particles activated into cloud droplets.

The nonlinearity of BC RF wrt. the magnitude of emissions has implications concerning studies where BC climate effects are evaluated. In such studies, higher-than-realistic BC emissions are commonly used, in order to obtain a robust signal for ERF or the climate response, in the presence of noise related to internal climate variability. For example, Flanner (2013) scaled present-day global BC emissions by a factor of 2, Stjern et al. (2017) by a factor of 10, and Sand et al. (2015) by a factor of 25, while Yang et al. (2019) considered regional scaling factors of 3.5–14 for midlatitude emissions and 75–150 for Arctic emissions. The decrease of SF with BC emissions in Fig. 8 suggests that the simplest approach to estimating the climate response to present-day BC emissions from very high-emission experiments (i.e., scaling down the climate response by the multiplication factor applied to BC emissions) would lead to an underestimate.

# 7    Conclusions

A large array of simulations was conducted with the NorESM1-Happi model to systematically evaluate how the instantaneous RF normalized by the mass of emitted BC depends on the location and season of emissions. By running the aerosol scheme in an offline mode, essentially noise-free estimates of RF were obtained. The global-mean values of the direct RF due to BC in air (dirRF) and the RF due to BC in snow/ice (snowRF) were analyzed and normalized by emissions to obtain the corresponding specific forcings, dirSF and snowSF. Overall, it was found that dirSF and, even more so, snowSF depends strongly on the location and timing of the emissions. More specifically:

– dirSF is generally largest for BC emissions in tropical convective regions and for subtropical and midlatitude continents in summer, both due to abundant solar radiation and due to convective transport, which increases the BC lifetime and the amount of BC above clouds. BC dirSF is also relatively large for emissions in high-albedo high-latitude regions such as Antarctica and Greenland. Even for seasonally uniform emissions, dirSF varies more than by a factor of ten depending on the emission location. The dependence on emission location is much stronger than suggested by previous studies that have considered emissions from continental/subcontinental-scale regions.

– For BC emissions at low latitudes, snowSF is generally negligible, while it can be very large for high-latitude emissions. The largest values of snowSF occur for BC emissions in polar regions, and they substantially exceed the maxima of dirSF found for low-latitude emissions. This, together with the high efficacy of snowRF found in previous work, strongly suggests that for a given mass of BC emitted, also the climate effects would be largest for high-latitude emissions. This notion was further supported by a rough analysis of the global-mean temperature response based on regional climate sensitivity coefficients derived in earlier studies.

– The longitude of BC emissions can also have a large impact on the SF. Although the largest dirSF is simulated for BC emissions in the Tropical Warm Pool, otherwise dirSF is mostly larger for emissions over land than over ocean, especially in summer. BC snowSF depends critically on how long snow persists in spring near/downwind of the emission region. For example, for the subarctic region, snowSF is substantially larger for BC emissions in Siberia and northern North America than for equally large emissions in Fennoscandia, due to the longer snow season and lesser vegetation masking effects in the former regions.

– The dependence of dirSF on emission season is especially large at high and middle latitudes because of the strong seasonal cycle of insolation. BC dirSF is smallest for emissions in local winter and usually largest for emissions in summer (or spring, for emissions in the Arctic and in the southern parts of the Southern Ocean). Over land, the impact of increased insolation in summer is reinforced by increased BC lifetime associated with lesser atmospheric stability and stronger convection.

– While the BC atmospheric lifetime is short (mostly one week or less), BC deposited on snow in the dark season can absorb solar radiation in spring as long as there is snow left. Thus, emissions in all seasons with snow contribute to

snowSF. The annual-mean snowSF is generally largest for BC emissions in spring in the northern hemisphere high latitudes, and for BC emissions in summer in Antarctica.

The additivity of the RFs resulting from BC emissions in different regions and seasons was also investigated. It was found that for current-day emissions, dirRF is almost additive: summing the RFs computed for individual regions/seasons without considering BC emissions from elsewhere overestimates dirRF by less than 10%. For snowRF, the overestimate is somewhat larger, ∼20%. The overestimation primarily arises from partial saturation of BC absorption with increasing BC burden in snow, and to a lesser extent, in air. The tendency towards partial saturation of absorption, and hence, the nonlinearity of BC RF wrt. the magnitude of emissions, becomes more pronounced for larger emissions. This should be taken into account when interpreting model experiments in which BC climate effects are evaluated, as such experiments often use higher-than-realistic emissions to improve the signal-to-noise ratio.

Concerning climate change mitigation policy, the present study strongly suggests that in the calculation of emission metrics for BC, not all BC emissions should be treated the same: rather, it would be important to account for the location and timing of the emissions. Yet, this is not a straightforward task. In this context, two key limitations of the current study have to be considered. First, this study is conducted with a single climate model, and is thus subject to model-specific biases. Second, and more fundamentally, we have focused on the instantaneous RF due to the interaction of BC in air and snow with solar radiation, ignoring rapid adjustments and BC indirect effects on clouds. While the GWP metric can be computed using the instantaneous RF, previous research indicates that for BC, the instantaneous RF (and hence GWP) is not a very good metric of the climate response. Deriving, for example, the Global Temperature Potential would require evaluating the temperature response in model runs in which BC is allowed to interactively influence the simulated climate. However, resolving the GTP as a function of emission region/season at a resolution comparable to that achieved for RF in the present work would be computationally unfeasible, due to signal-to-noise considerations. Therefore, in order to derive a rough estimate of the global annual-mean temperature response associated with BC emissions from different regions, we employed regional climate sensitivity coefficients that link the global or regional temperature response $\Delta T$ to the RF in a given region (Shindell and Faluvegi, 2009; Collins et al., 2013; Flanner, 2013; Sand et al., 2016). This analysis however involves substantial uncertainties, not least due to the very coarse resolution at which the RCSs were defined (four wide latitude bands).

The above limitations notwithstanding, this paper provides a current best guess of the impacts of regional BC emissions on global radiative forcing and temperature. To facilitate the use of these results, a fortran-based tool is provided that computes the radiative forcings and associated temperature response based on the region and annual or monthly BC emission rates given by the user (Räisänen, 2022a). This tool could potentially be helpful for applications such as computation of climate metrics for BC or evaluation of climate effects of BC emissions in integrated assessment models.

*Code and data availability.* The NorESM1-Happi model code is available for registered users. To register for access, users should contact noresm-ncc@met.no and briefly state the purpose of the use of the model and sign a user agreement. Code modifications of NorESM1-Happi employed in this work can be obtained by contacting the first author. Scripts, computer programs and climate model data used for producing

the results discussed in this manuscript are available at https://doi.org/10.23728/FMI-B2SHARE.37E9742A67B8415E8F8102A8D9757F6B (Räisänen, 2022b). A fortran-based tool that computes the radiative forcings and associated temperature response based on the region and annual or monthly BC emission rates is available at https://doi.org/10.5281/zenodo.6461647 (Räisänen, 2022a).

## Appendix A: Diagnostic shortwave radiation calculations

The RF calculations in Sect. 2.2 involve four net shortwave fluxes $F_{\text{air+snow}}$, $F_{\text{air}}$, $F_{\text{snow}}$, and $F_{\text{aie}}$. Each of them is derived from a separate diagnostic call to the model shortwave radiation scheme. For $F_{\text{aie}}$, there is also a separate call to the cloud microphysics scheme (Rasch and Kristjánsson, 1998).

- $F_{\text{air+snow}}$ is the net SW flux in the case in which the BC simulated by NorESM1-Happi's aerosol scheme is taken into account both in the computation of optical properties of atmospheric aerosols and snow albedo. Cloud properties are
600 determined using CCSM4's prescribed aerosols and cloud droplet number concentrations, so they are not influenced by the simulated BC.

- $F_{\text{air}}$ is calculated otherwise as $F_{\text{air+snow}}$ but BC concentration is set to zero in the snow albedo calculation.

- $F_{\text{snow}}$ is calculated otherwise as $F_{\text{air+snow}}$ but BC concentration is set to zero in the calculation of atmospheric aerosol optical properties.

- $F_{\text{aie}}$ is calculated for evaluating the aerosol indirect effect on liquid-phase clouds. As a difference to the other calls to the shortwave scheme, the values of cloud droplet effective radius and liquid water content used in this call are derived using NorESM1-Happi's interactive aerosols in the cloud microphysics scheme (Kirkevåg, 2013). In order to eliminate the contribution from BC direct radiative effects and the effect on snow albedo, aerosol optical properties are based on CCSM4's prescribed standard aerosols and BC concentration in snow is set to zero.

Finally, after the diagnostic calls, the shortwave radiation scheme is invoked one more time, for computing the radiative fluxes used for advancing the model state. In this computation, both the cloud and aerosol properties are based on CCSM4's prescribed standard aerosols while the effect of BC on snow albedo is neglected. This ensures that the model state remains independent of the BC emissions employed by NorESM1-Happi's aerosol scheme.

## Appendix B: Methods used for estimating the global-mean temperature response

We employ regional climate sensitivity coefficients to estimate the global annual-mean temperature response $\Delta T_{\text{glob}}$ and the corresponding temperature response normalized by BC emissions $\Delta T_{\text{glob,norm}}$. The RCSs were originally computed with the GISS model (Shindell and Faluvegi, 2009; Collins et al., 2013) but augmented by CESM1.0.3 simulations (Flanner, 2013) for the Arctic climate response to local BC RF in the Arctic. Our approach follows that employed by Sand et al. (2016) to estimate the Arctic temperature response but extends it to the computation of $\Delta T_{\text{glob}}$.

To produce the results in Fig. 7c,d, $\Delta T_{\mathrm{glob}}$ was estimated for the experiments with seasonally uniform emissions in the 192 emission regions. For each of these experiments, averages of dirRF and snowRF were calculated over the four wide latitude bands considered by Shindell and Faluvegi (2009): 90–28°S, 28°S–28°N, 28–60°N, and 60–90°N. Then, $\Delta T_{\mathrm{glob}}$ was evaluated as

$$\Delta T_{\mathrm{glob}} = \sum_{l=1}^{4} w_l \sum_{m=1}^{4} \left[ \mathrm{RCS}_{\mathrm{dirRF},l,m} \mathrm{dirRF}_m + \mathrm{RCS}_{\mathrm{snowRF},l,m} \mathrm{snowRF}_m \right] \tag{B1}$$

Here, $w_l$ is the areal fraction of latitude band $l$, $\mathrm{dirRF}_m$ and $\mathrm{snowRF}_m$ are averages over latitude band $m$, and $\mathrm{RCS}_{\mathrm{dirRF},l,m}$ and $\mathrm{RCS}_{\mathrm{snowRF},l,m}$ quantify the temperature response in latitude band $l$ to BC radiative forcing in band $m$, separately for dirRF and snowRF.

The RCSs employed in Eq. (B1) are given in Tables B1 and B2. The $\mathrm{RCS}_{\mathrm{dirRF}}$ coefficients are mainly taken from Table 3 of Collins et al. (2013), which is based on experiments made with the GISS-ER model by Shindell and Faluvegi (2009). As an exception, CESM1.0.3 experiments made by Flanner (2013) are utilized to estimate the Arctic temperature response to local BC dirRF in the Arctic based on the vertical profile of the additional shortwave absorption $\Delta \mathrm{SWABS}_k$ caused by BC in each model layer $k$:

$$\mathrm{RCS}_{\mathrm{dirRF}} = 2.9/4.0 \times \frac{\sum_{k=1}^{n} \Delta \mathrm{SWABS}_k \mathrm{RCS}_{\mathrm{dirRF,VR,k}}}{\sum_{k=1}^{n} \Delta \mathrm{SWABS}_k} \tag{B2}$$

The vertically-resolved regional climate sensitivity coefficients $\mathrm{RCS}_{\mathrm{dirRF,VR,k}}$ are based on a fit to the simulation results by Flanner (2013), previously used by Sand et al. (2016) (see Fig. B1). Following Sand et al. (2016), the values are scaled by the ratio of GISS and CESM climate sensitivities (2.9 K vs. 4.0 K). The RCS coefficient to local BC snowRF in the Arctic is likewise based on CESM1.0.3 simulations by Flanner (2013) (and again scaled by 2.9/4.0 as in Sand et al., 2016). Otherwise it is assumed based on earlier research (Hansen et al., 2005; Flanner et al., 2007) that $\mathrm{RCS}_{\mathrm{snowRF},l,m} = 3\mathrm{RCS}_{\mathrm{dirRF},l,m}$.

Due to several reasons, the temperature responses computed using Eq. (B1) must be regarded as very rough estimates. First, the RCSs come from other climate models than that used here for computing the RFs. Second, the very coarse spatial resolution of the RCSs might cause some artificial features. For example, the same RCSs are used from the South Pole to the poleward edge of the southern subtropics (90–28°S). Third, since Shindell and Faluvegi (2009) did not actually perform experiments addressing the role of BC radiative forcing between 90–28°S, the RCSs for forcing in this latitude band are taken from their $CO_2$ experiments (Table 3 in Collins et al., 2013). Fourth, because Shindell and Faluvegi (2009) did not evaluate $\mathrm{RCS}_{\mathrm{snowRF},l,m}$, we follow the approach of Sand et al. (2016) and assume $\mathrm{RCS}_{\mathrm{snowRF},l,m} = 3\mathrm{RCS}_{\mathrm{dirRF},l,m}$ based on Hansen et al. (2005) and Flanner et al. (2007), except for the Arctic temperature response to the local snowRF in the Arctic (Flanner, 2013). Fifth, the vertical profile of BC dirRF is taken into account only when computing the Arctic temperature response to the local dirRF in the Arctic (Flanner, 2013; Sand et al., 2016), although previous research suggests that the vertical profile is also relevant at lower latitudes (Hansen et al., 2005; Ban-Weiss et al., 2012).

Finally, the estimates of the contribution of global BC emissions to the global-mean temperature $\Delta T_{\mathrm{glob}}$ (in the last paragraph of Sect. 6.2) were produced as follows. First, we derived estimates of the normalized temperature response $\Delta T_{\mathrm{glob,norm}}$ to BC emissions in the 192 lat-lon boxes similar to that shown in Fig. 7c but separately for emissions in each season. This entails the

assumption that the RCSs are independent of the seasonal distribution of emissions and RF. While this assumption is unverified and not necessarily accurate, we are not aware of RCSs computed for seasonal emissions. Second, $\Delta T_{\text{glob}}$ corresponding to

the actual distribution of emissions was computed as a sum over the four seasons and the 192 lat-lon boxes:

$$\Delta T_{\text{glob}} = \sum_{s=1}^{4} f_{\text{time},s} \sum_{i=1}^{12} \sum_{j=1}^{16} A_{i,j} \epsilon_{ijs} \Delta T_{\text{glob,norm},ijs} \tag{B3}$$

Here, $\epsilon_{ijs}$ is the emission rate averaged over the season $s$ and the lat-lon box with longitude index $i$ and latitude index $j$, $f_{\text{time},s}$ is the time fraction for season $s$ (close to 0.25), $A_{i,j}$ is the lat-lon box area, and $\Delta T_{\text{glob,norm},ijs}$ is the global-mean temperature response normalized by emissions (in units $\text{K}(\text{kg}\,\text{s}^{-1})^{-1}$) for the lat-lon box $(i,j)$ and season $s$.

**Table B1.** Regional climate sensitivity coefficients employed in Eq. (B1) for BC direct radiative forcing ($\text{K}\,(\text{Wm}^{-2})^{-1}$).

|  |  | Forcing region | | | |
|---|---|---|---|---|---|
|  |  | 90°S–28°S | 28°S–28°N | 28°N–60°N | 60°N–90°N |
|  | 90°S–28°S | 0.19 | 0.06 | 0.02 | 0.00 |
| Response | 28°S–28°N | 0.09 | 0.17 | 0.07 | 0.02 |
| region | 28°N–60°N | 0.07 | 0.24 | 0.14 | 0.08 |
|  | 60°N–90°N | 0.06 | 0.31 | 0.15 | VR |

VR indicates the use of vertically-resolved coefficients; see Fig. B1.

**Table B2.** Regional climate sensitivity coefficients employed in Eq. (B1) for radiative forcing due to BC in snow ($\text{K}\,(\text{Wm}^{-2})^{-1}$).

|  |  | Forcing region | | | |
|---|---|---|---|---|---|
|  |  | 90°S–28°S | 28°S–28°N | 28°N–60°N | 60°N–90°N |
|  | 90°S–28°S | 0.57 | 0.18 | 0.06 | 0.00 |
| Response | 28°S–28°N | 0.27 | 0.51 | 0.21 | 0.06 |
| region | 28°N–60°N | 0.21 | 0.72 | 0.42 | 0.24 |
|  | 60°N–90°N | 0.18 | 0.93 | 0.45 | 1.06 |

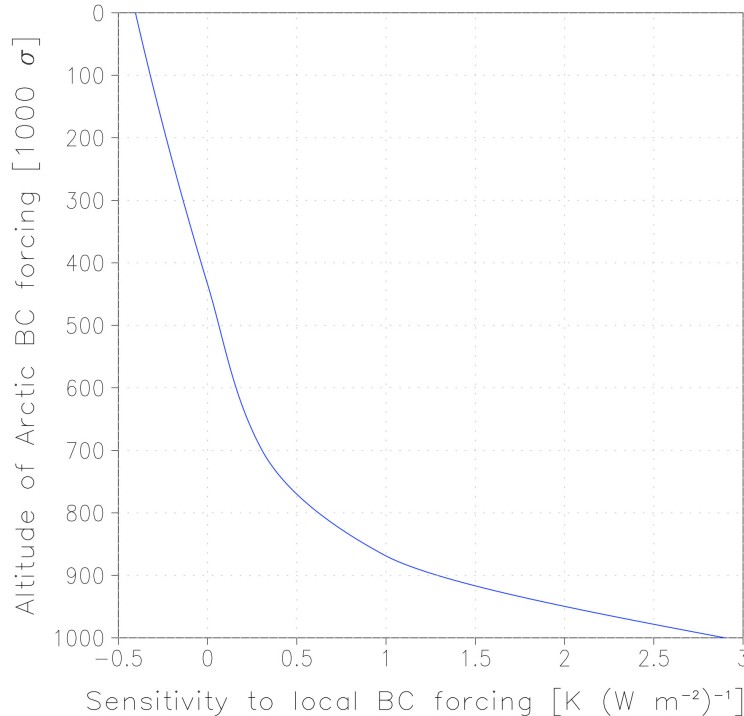

**Figure B1.** Vertically resolved regional climate sensitivity coefficients describing the Arctic surface temperature response to local BC direct radiative forcing in the Arctic. $\sigma$ equals the local air pressure normalized by the surface pressure. This is a fit to climate model experiments by Flanner (2013), originally presented in Fig. S2 of Sand et al. (2016).

*Author contributions.* PR conceived the idea, run and analyzed the NorESM1-Happi experiments, and wrote the first manuscript draft. AK and ØS provided scientific and technical advice on NorESM1-Happi. All authors commented on the manuscript.

*Competing interests.* The authors declare no competing interests.

*Acknowledgements.* PR, JM and RM acknowledge funding from the EU Framework Programme for Research and Innovation Horizon 2020 under grant agreement no. 101003826 (CRiceS) and funding by the Nordic Council of Ministers through the NorFORCeS project (grant
no. NKL-2011). PR was also partially funded by the Academy of Finland through the NABCEA and SnowAPP projects (decision numbers 296302 and 315497). The work of M. Savolahti was partially funded by the Academy of Finland through the NABCEA and BBrCaC projects (decision numbers 296302 and 341949), and by the Black Carbon Footprint project financed by Business Finland (grant nr:1466/31/2019, 530/31/2019, 528/31/2019, 1462/31/2019). M. Sand has been supported by the Research Council of Norway (grant no. 315195, ACCEPT). NorESM1-Happi has been developed with contributions from the Research Council of Norway (grant no. 261821, Happi-Eva) and the

EU Framework Programme for Research and Innovation Horizon 2020 (grant no. 641816, Crescendo). Kaarle Kupiainen and Ville-Veikko Paunu are thanked for help with the ECLIPSE V6b CLE dataset. The figures in this article have been produced using the GrADS software (http://cola.gmu.edu/grads/) and Scientific colour maps by Crameri et al. (2020) were used in many of them.

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
