# Peer review of "Mapping the dependence of BC radiative forcing on emission region and season"

_Atmospheric Chemistry and Physics, 2022_

## Author Response (AR1)

We thank Anonymous Referee #1 and Prof. William Collins for their constructive comments on the manuscript. Point-by-point responses to the comments are provided below. The referee comments are written in *italic* font and our responses in normal font. The comment numbers have been added by the authors. The line numbers given in red font mainly refer to the marked-up manuscript. In some cases (i.e., at places where a combination of additions and deletions makes the marked-up manuscript messy to read), line numbers in the revised manuscript are also provided. At each occasion, it is explicitly stated which manuscript version is referred to.

**Responses to comments by Anonymous Referee #1**

**Comment:** *1. Limitations of the present study (which are discussed in Section 6.2) include the fact the authors are not able to quantify the rapid adjustments, which have been shown to be very important for the climate impacts (e.g., Stjern et al. 2017). See also*

*Smith, C. J., Kramer, R. J., Myhre, G., Forster, P. M., Soden, B. J., Andrews, T., et al. (2018). Understanding rapid adjustments to diverse forcing agents. Geophysical Research Letters, 45, 12,023– 12,031. https://doi.org/10.1029/2018GL079826*

**Response:** We have added a bit more discussion about the rapid adjustments in the beginning of Section 6.2 of the revised manuscript, also citing the paper by Smith et al. (2018).

**Change in the manuscript**: Modified version of the first paragraph of Sect. 6.2 (lines 431–445 in the marked-up manuscript, 409–421 in the revised manuscript): "This study has focused on the instantaneous RF of BC only, including dirRF and snowRF. The problem with this is that for BC, the instantaneous RF is, in fact, not a very good predictor of the ensuing climate response. Previous studies have shown that the climate response to BC dirRF may depend strongly on the spatial distribution of BC, and especially its vertical profile (Hansen et al. 2005, Ban-Weiss et al. 2012, Flanner, 2013, Ocko et al. 2014, Samset and Myhrew 2015). In particular, Samset and Myhre (2015) demonstrated that while dirRF is largest when BC is located at high altitudes, the temperature response is largest for BC close to the surface. The disconnection between temperature response and dirRF is related to rapid adjustments in atmospheric stability, humidity and cloudiness. In general, for a realistic global distribution of BC, these adjustments act to make the BC effective radiative forcing (ERF) smaller than the instantaneous dirRF, thereby leading to a reduced global-mean temperature response (Hansen et al. 2005, Stjern et al. 2017, Smith et al. 2018, Richardson et al. 2019). This translates into low efficacy (the ratio of global-mean temperature responses to BC and $CO_2$ for a given RF) when defined wrt. the instantaneous or stratospherically adjusted RF (Hansen et al. 2005, Richardson et al. 2019). In contrast,

high efficacy has been reported for BC snowRF (Hansen and Nazarenko 2004, Hansen et al. 2005, Flanner et al. 2007) due to albedo feedbacks associated with accelerated snowmelt and due to warming concentrated near the surface."

**Comment:** *2. L130 "the semidirect effect of BC cannot be included". Shouldn't this be more general, i.e., rapid adjustments cannot be included? Semi-direct effects traditionally refer to clouds alone, but there are several rapid adjustments including those associated with the clouds.*

**Response and change in the manuscript:** We agree. The term "rapid adjustments" is used in the revised manuscript, in this specific case as well as elsewhere (lines 68, 137, 438, 452, 467 and 598 in the marked-up manuscript).

**Comment:** *3. Section 2.3: In the context of BC emissions, the two main sources are fossil fuel and biomass burning. In contrast to fossil fuel BC emissions, biomass burning BC emissions are likely less easily controlled to mitigation policies. Is there any utility in separating the two? Probably beyond the scope of this work, but perhaps the authors could comment.*

**Response:** In principle, biomass burning BC emissions are also relevant for mitigation considerations, because they are partly anthropogenic. For example, a large fraction of forest fires are caused by humans, either intentionally or unintentionally. At any rate, from the point of view of aerosol modeling, the reason for treating fossil-fuel and biomass burning BC emissions separately in the aerosol scheme is that BC particles from these sources are processed somewhat differently in the atmosphere (e.g., biomass burning BC is typically emitted at higher altitudes than fossil-fuel BC, the particles are larger, and they experience different mixing processes with other aerosols).

**Change in the manuscript**: We agree with Anonymous Referee #1 that the question of mitigation of fossil-fuel vs. biomass burning aerosols is beyond the scope of our paper. Therefore, no change is made to the manuscript.

**Comment:** *4. L200. In the context of convective lofting, see also:*

*Park, S., and Allen, R. J. (2015), Understanding influences of convective transport and removal processes on aerosol vertical distribution, Geophys. Res. Lett., 42, 10,438– 10,444, doi:10.1002/2015GL066175.*

**Response and change in the manuscript**: A reference to Park and Allen (2015) has been

included in the revised manuscript (line 219 in the marked-up manuscript). The paper by Sand et al. (2015) (already included in the reference list of the original manuscript) is also relevant in this context, so it is cited too.

**Responses to comments by William Collins**

**Comment:** *1. I understand the reluctance for detailed indirRF calculations since they are not likely to be additive, but it might still be useful to see the regional variation. A version of fig 4 for indirRF could be added to the supplement. Why is the indirRF positive? Does mixing with BC reduce the nucleating ability of SO4, or does it reduce the SO4 lifetime? How is indirRF calculated? Is it a double call as in Ghan 2013?*

**Response:** To start with, the calculation of indirRF is based on an extra diagnostic call to the shortwave radiation scheme (after a respective extra call to the cloud microphysics scheme), but it differs from Ghan (2013). Firstly, a diagnostic shortwave net radiative flux $F_{\text{aie}}$ is computed in each experiment so that

- The cloud droplet effective radius and liquid water content are evaluated using BC and other aerosols simulated by NorESM1-Happi's interactive aerosol scheme. This requires a separate call to the cloud microphysics scheme (Rasch and Kristjansson 1998, https://doi.org/10.1175/1520-0442(1998)011<1587:ACOTCM>2.0.CO;2).

- To eliminate the contribution from the aerosol direct radiative effects and the effect on snow albedo, aerosol optical properties used in computing $F_{\text{aie}}$ are based on CCSM4's prescribed standard aerosols (which are the same in all runs) and BC concentration in snow is set to zero.

Finally, the indirRF due to BC is defined as the difference of $F_{\text{aie}}$ to the case with no BC aerosols:

$$\text{indirRF} = F_{\text{aie}} - F_{\text{aie},0} \tag{1}$$

There is similarly also a change in the longwave fluxes (a LW IndirRF), arising from changes in cloud emissivity as the liquid the water content is modified (from the extra cloud microphysics call using NorESM-Happi aerosols). This effect is however very small (about 0.002 W m$^{-2}$ in the global mean for the experiment REAL, i.e. nearly two orders of magnitude smaller than the shortwave indirRF), and it has been ignored throughout this paper.

Figure 1 shows the indirect specific forcing (indirSF) calculated as a function of emission region. We do not consider it quantitatively meaningful, due to the strong nonlinearities, but it can still be added to the Supplementary material. Perhaps the message here is

that irrespective of the emission region, the resulting indirSF is positive, and especially so for low-latitude oceans.

[Figure]

Figure 1: Global-mean values of BC indirect specific forcing (TJ kg$^{-1}$) for the experiments in which uniform BC emissions over the year were assumed, separately for emissions in each of the 192 lat-lon boxes.

[Figure]

Figure 2: Global-mean vertical profiles of (a) CDNC, (b) cloud liquid water, and (c) maximum superasaturation normalized by (d) warm cloud frequency of occurrence in the experiments without BC (no_BC, black) and with near-present BC emissions (REAL, red). In fact, warm cloud fraction would be smaller than the warm cloud frequency of occurrence, so the plotted profiles are somewhere between in-cloud and grid-mean values.

While the question of why the indirRF is positive is intriguing, answering it with certainty would require dedicated analysis, which is, in our opinion, beyond the scope of our paper. However, based on a couple of extra experiments, we were able to formulate a reasonable hypothesis about this matter. A brief form of this explanation has been added to Section 6.3. Here we outline this hypothesis in some more detail.

- The positive BC indirRF ($0.168$ W m$^{-2}$ in experiment REAL) is associated with slightly reduced cloud droplet number concentration CDNC (on average by 3.0%), marginally reduced cloud liquid water (by 0.48%), and a slight decrease in the maximum supersaturation diagnosed by the aerosol activation scheme.

- Two extra experiments were conducted, one including only fossil-fuel BC and another including only biomass burning BC. These types of BC are treated differently in the NorESM1-Happi aerosol scheme (see Fig. 1 in Kirkevåg et al., GMD2013, cited in the manuscript). The fossil-fuel BC particles are mainly injected to the nucleation mode (i.e., they are very small) and they subsequently get internally mixed with SO$_4$ from condensation of H$_2$SO$_4$. In contrast, biomass burning BC particles are larger (injected to the Aitken mode) and they are internally mixed with organic matter (and subsequently also with sulfate).

- The extra experiments revealed that the positive indirRF, the reduced CDNC and cloud liquid water content as well as the slightly reduced supersaturation are associated with BC particles from fossil-fuel emissions. Specifically, for the experiment with fossil-fuel aerosols only, the global-mean indirRF was slightly larger than in REAL ($0.183$ W m$^{-2}$), the CDNC was reduced on average by 3.4%, and cloud liquid water by 0.50%. Maximum supersaturation was also reduced essentially by the same amount as in the experiment REAL. In contrast, for biomass burning BC, indirRF was marginally negative ($-0.015$ W m$^{-2}$), CDNC and cloud liquid water were marginally increased (by 0.36% and 0.02%, respectively), and supersaturation was almost unchanged. This suggests that some of the biomass burning BC particles become activated as cloud droplets, causing a negative indirRF, but their number is far too small to compensate for the reduced CDNC and positive indirRF associated with fossil-fuel BC.

- Why then is indirRF positive for fossil-fuel BC? A plausible explanation is that fossil-fuel BC in NorESM1-Happi acts as a condensation sink for both sulfuric acid and water vapor while not being large enough to activate into cloud droplets. This is exacerbated by the fossil-fuel BC particles being very small in NorESM1-Happi (90% of fossil-fuel BC mass is emitted to the nucleation mode with a diameter 23.6 nm and 10% to the fractal agglomerate mode with a diameter 200 nm; while in other models fossil-fuel BC is emitted at sizes between 30 nm and 100 nm (Mann et al. 2014;

doi:10.5194/acp-14-4679-2014)). When numerous small fossil-fuel BC particles become coated with $SO_4$, they reduce the amount of condensable sulfuric acid available for the growth of aerosols into activation sizes, and they also become hygroscopic and act as a sink for water vapor. Therefore, they can both inhibit the formation of large activation-size aerosol particles and reduce slightly the maximum supersaturation diagnosed by the Abdul-Razzak and Ghan (2000) scheme (see Fig. 2c) which then further reduces the number of aerosol particles activated into cloud droplets.

**Changes in the manuscript**: The above equation for indirRF has been added to section 2.2 (lines 146–153 in the marked-up manuscript). The details regarding the computation of $F_{aie}$ are included in the new Appendix A ("Diagnostic shortwave radiation calculations"), which also explains how the other radiative fluxes used in evaluating the RFs (i.e., $F_{air+snow}$, $F_{air}$, and $F_{snow}$ in Eqs. (1) and (2) of the paper) are computed. Please see lines 619–638 in the marked-up manuscript.

The above Fig. 1 has been added to the Supplementary (Fig. S7 in the revised version).

Regarding the possible reason for the positive indirRF, the following text has been added in Section 6.3 (lines 534–542 in the marked-up manuscript): "A possible explanation for the positive indirRF in NorESM1-Happi is that fossil-fuel BC can actually reduce the cloud droplet number concentration by acting as a condensation sink for sulfuric acid and water vapor while not being large enough to activate into cloud droplets (Koch et al. 2011, Blichner et al. 2021). In NorESM1-Happi, 90% of fossil-fuel BC mass is emitted to the nucleation mode with a diameter 23.6 nm and 10% to the fractal agglomerate mode with a diameter 200 nm, while in other models fossil-fuel BC is emitted at sizes between 30 nm and 100 nm (Mann et al. 2014). When numerous small fossil-fuel BC particles become coated with $SO_4$, they reduce the amount of condensable sulfuric acid available for the growth of aerosols into activation sizes, and they also become hygroscopic and act as a sink for water vapor. Thus, they can both inhibit the formation of large activation-size aerosol particles and reduce slightly the maximum supersaturation, which then further reduces the number of aerosol particles activated into cloud droplets."

**Comment:** *2. Even though they can't be addressed in this study, there could be a bit more mention of meteorological adjustments to BC, for instance the increased stabilisation of the atmospheric profile, and how they might affect the conclusions. Stjern et al. is cited, but not the discussions in that paper, also there are Samset papers. These meteorological adjustments will be included implicitly in the Shindell and Faluvegi coefficients. What they term "efficacy" is really an accounting for adjustments.*

**Response:** Efficacy indeed depends on which radiative forcing is used in its definition, and the low efficacy pertains to the case when the efficacy is defined wrt. the instantaneous RF

(or the stratosphere-adjusted RF, which is nearly the same for BC) but indeed not necessarily when it is defined wrt. ERF. We have modified the first paragraph of Sect. 6.2. so as to explicitly mention the role of rapid adjustments and the underlying physics. A few more references have also been added (Samset and Myhre JGR 2015, Smith et al. GRL 2018, Richardson et al. JGR 2019).

**Change in the manuscript**: Modified version of the first paragraph of Sect. 6.2 (lines 431–445 in the marked-up manuscript, 409–421 in the revised manuscript): "This study has focused on the instantaneous RF of BC only, including dirRF and snowRF. The problem with this is that for BC, the instantaneous RF is, in fact, not a very good predictor of the ensuing climate response. Previous studies have shown that the climate response to BC dirRF may depend strongly on the spatial distribution of BC, and especially its vertical profile (Hansen et al. 2005, Ban-Weiss et al. 2012, Flanner, 2013, Ocko et al. 2014, Samset and Myhre 2015). In particular, Samset and Myhre (2015) demonstrated that while dirRF is largest when BC is located at high altitudes, the temperature response is largest for BC close to the surface. The disconnection between temperature response and dirRF is related to rapid adjustments in atmospheric stability, humidity and cloudiness. In general, for a realistic global distribution of BC, these adjustments act to make the BC effective radiative forcing (ERF) smaller than the instantaneous RF, thereby leading to a reduced global-mean temperature response (Hansen et al. 2005, Stjern et al. 2017, Smith et al. 2018, Richardson et al. 2019). This translates into low efficacy (the ratio of global-mean temperature responses to BC and $CO_2$ for a given RF) when defined wrt. the instantaneous or stratospherically adjusted RF (Hansen et al. 2005, Richardson et al. 2019). In contrast, high efficacy has been reported for BC snowRF (Hansen and Nazarenko 2004, Hansen et al. 2005, Flanner et al. 2007) due to albedo feedbacks associated with accelerated snowmelt and due to warming concentrated near the surface".

**Comment:** *3. Introduction: Could also cite Aamaas et al. 2016 and Bellouin et al. 2016 papers from ECLIPSE.*

**Response:** A paragraph discussing Bellouin et al. (2016) has been added to the introduction, but we leave out Aamaas et al. (2016), because it does not bring new information on BC radiative forcing, compared to Bellouin et al. (2016). Rather, their paper uses the Bellouin et al. (2016) radiative forcings to derive more elaborate emission metrics. Aamaas et al. (2016, 2017) will be discussed later in Section 6.2, though.

**Change in the manuscript**: The following text has been added to the Introduction (lines 64–69 in the marked-up manuscript): "Bellouin et al. (2016) compared, as a part of the multi-model ECLIPSE study, the impact of reducing BC emissions from Europe and East

Asia. They found, for the sum of dirRF and the indirect RF (indirRF) due to aerosol-cloud interaction, that the RF normalized by emitted mass was larger for BC emission perturbations in summer than in winter, and slightly larger for European than East Asian emissions. They also reported (although based on a single model only) a negative RF contribution due to rapid adjustments in summer and a substantial positive snowRF especially for European BC emissions in winter, both of these factors counteracting the seasonal variation associated with the sum of dirRF and indirRF."

**Comment:** *4. Line 85: This should mention the magnitude of indirRF here - it seems to be 25% of dirRF.*

**Response and change in the manuscript**: The following sentence has been added (line 93 in the marked-up manuscript): "In fact, for NorESM1-Happi, it amounts to roughly 25% of dirRF."

**Comment:** *5. Line 130: Suggest to use "meteorological adjustments" or "rapid adjustments" rather than "semi-direct effect" to align with IPCC terminology.*

**Response and change in the manuscript:** We agree. The term "rapid adjustments" is used in the revised manuscript (lines 68, 137, 438, 452, 467 and 598 in the marked-up manuscript).

**Comment:** *6. Line 136: Give some explanation of how F_air is calculated. Presumably this is from a double call to the radiation scheme with zero BC in the advancing step.*

**Response:** $F_{air}$ is the net solar radiative flux calculated when BC is included only in air (i.e., not in snow). This is a diagnostic radiation calculation (i.e., an extra call to the radiation scheme). The model advancing step in fact includes BC in air; however, it uses CCSM4's prescribed standard aerosols. This information was in principle already available in the original manuscript (see lines 121–122, 134–135 and 138–139), but we have tried to make it clearer in the revised version.

**Change in the manuscript**: The details regarding the computation of $F_{air}$ as well as the other shortwave radiative fluxes used for deriving radiative forcings have been included in the new Appendix A ("Diagnostic shortwave radiation calculations"; lines 619–638 in the marked-up manuscript). Specifically, it is stated on lines 623–627 that "$F_{air+snow}$ is the net SW flux in the case in which the BC simulated by NorESM1-Happi's aerosol scheme is taken into account both in the computation of optical properties of atmospheric aerosols and snow albedo. Cloud properties are determined using CCSM4's prescribed standard

aerosols, so they are not influenced by the simulated BC. $F_{\mathrm{air}}$ is calculated otherwise as $F_{\mathrm{air+snow}}$ but BC concentration is set to zero in the snow albedo calculation."

**Comment:** *7. Section 2.3: This might comment on how the mixing affects other species. Does mixing with SO4 affect the lifetime of BC, i.e. do BC emissions have a lower lifetime if they are emitted in a high SO4 region? Does mixing with BC affect the lifetime of SO4, i.e. is there an indirect effect of BC on SO4 RF and if so is this included in F_air?*

**Response:** We found experimentally that the impact of BC on $SO_4$ burden and thus $SO_4$ lifetime is extremely small (e.g., for experiment REAL the global-mean $SO_4$ burden differed only 0.02% from the case with no BC). Likewise, the impact of $SO_4$ on BC lifetime appeared extremely small (e.g., cutting all anthropogenic $SO_2$ emissions by 90% reduced the global-mean $SO_4$ burden by 51% but BC burden by merely 0.004%). Similarly, the impacts of BC emissions on particulate organic matter (and vice versa) were found to be non-zero but negligibly small. Why this is so, and whether that is fully realistic, might be interesting questions as such, but they fall outside the scope of our paper.

To the extent that BC changes the concentration of other aerosols, the effects are implicitly included in the diagnostic radiative fluxes ($F_{\mathrm{air+snow}}$, $F_{\mathrm{air}}$, and $F_{\mathrm{snow}}$) — which actually means that they cancel out when computing dirRF and snowRF using Eqs. (1) and (2) in the manuscript. But everything suggests that these effects are very minor.

**Change in the manuscript**: The following has been added to the end of Section 2.3 (lines 168–169 in the marked-up manuscript): "Sensitivity tests indicated that the impacts of $SO_2$ and OM emissions on the BC burden simulated by NorESM1-Happi are very small. Likewise, BC emissions have very little impact on the burdens of $SO_4$ and OM."

**Comment:** *8. Figure 1. I suggest splitting into COARSE-REAL which shows the impact of resolution, and then RECONST-REAL which shows the additivity. I'm not sure RECONST-REAL is that useful since it mixes these two effects.* Presumably there is a typo here: the first RECONST−REAL should be RECONST−COARSE?

**Response:** We chose to keep RECONST−REAL, to show directly the error that results from the application of the reconstuction formula to realistic emissions. But for the physical understanding of the results, it is indeed a good idea to show explicitly the error associated with misrepresented emissions (i.e., COARSE−REAL) and the additivity error (RECONST−COARSE). So in the revised version, Figures 1 and 2 (as well as S1–S5) show the following:

- RECONST−REAL: the reconstruction error for the REAL experiment

- COARSE−REAL: the "emission error" resulting from the the coarse resolution of emissions, as well as from the treatment of all BC as fossil-fuel BC.

- RECONST−COARSE: the "additivity error" (which equals the reconstruction error for the COARSE experiment)

**Change in the manuscript**: New panels showing explicitly the emission error (COARSE−REAL) have been added to Figs.1 and 2 (and Figs. S1–S5 in the Supplementary material). The figure captions and the text in Sect. 4.2 have been modified accordingly (lines 234–270 in the marked-up manuscript, lines 224–249 in the revised manuscript).

**Comment:** *9. Line 240: Why aren't the reconstructed fields for REAL and COARSE actually identical as opposed to "virtually" identical. Similarly, in fig S1 are the emissions for RECONST and COARSE identical, and if not, why not?*

**Response:** You are right, they should be identical! The reason for why they are not strictly identical is that the emissions output by the model actually differ slighly from the emissions given as input. On average, the output emissions are smaller by ca. 1%, with slight geographic variations in the difference. The root cause of this difference is currently unknown, but it seems to be an issue with the model diagnostics. At any rate, in our opinion, anything that causes an uncertainty of the order of 1% in the simulation of BC or its radiative forcing in a climate model is practically insignificant in the face of other uncertainties (realistically, the model biases might be several tens of percent or even more).

In the original manuscript version, the reconstruction (Eq. 3) applied the emissions output by the model and it was therefore influenced by the issue noted above. Specifically, the reconstructed emissions for COARSE were on average 0.13% smaller than for REAL, with local differences varying from −0.88 to 0.45%, and the spatial correlation between the reconstructed fields was 0.999993. For the other quantities considered in Figs. 1, 2, and S2–S5, the differences were similar or even smaller.

Although the practical significance of this discrepancy is minimal, we chose to eliminate it in the revised version. This was achieved by using the model input emissions instead of the ouput emissions.

**Change in the manuscript**: The reconstructed values were recalculated, as noted above. As expected, the results are practically unchanged. There is one "macroscopic" change in the numerical values (one correlation dropped from 0.965 to 0.952; line 235 in the marked-up manuscript) but this is not due to the changed treatment of emissions (rather, the value in the original version was picked from a wrong line in a table).

**Comment:** *10. Figure 3: Why doesn't dirRF scale with column burden? I would have expected 3(a) and 3(c) to be much more similar since section 6.3 suggests little non-linearity in dirRF. A plot of dirRF/columnBC would be useful in the supplement.*

**Response:** There is no contradiction here. The results in Section 6.3 show that *when the spatiotemporal pattern of emissions is kept the same*, but the magnitude of emissions scaled, the ratio of global-mean dirRF to global-mean columnBC depends rather weakly on the magnitude of emissions. This does not rule out the possibility that the *local ratio of dirRF/columnBC* varies significantly as a function of geographical location. It is well-known from previous research (and physically intuitive) that in addition to BC burden, dirRF is influenced by other factors like the albedo of underlying surface, whether BC is above or below clouds, and the availability of solar radiation. In fact, our Figure 5b also points to this, as the ratio of the global mean dirSF to BC lifetime depends substantially on emission location, for the idealized experiments with emissions confined to a single region.

At any rate, we plotted the ratio of dirDF to burden corresponding to the experiment considered in Fig. 3.

**Change in the manuscript**: The following figure has been added to the Supplementary material (new Fig. S6), with some physical interpretation included in the figure caption.

[Figure]

Figure 3: Ratio of BC direct radiative forcing to BC burden ($MW\,kg^{-1}$) for the experiment in which a constant BC emission rate of $10^{-12}\,kg\,m^{-2}\,s^{-1}$ was applied in the lat-lon box 56.84–68.21°N, 1.25–31.25 °E (shown with a rectancle). Regions with very small BC burden (below $10^{-10}\,kg\,m^{-2}$) are screened out. The global mean value is indicated in the figure title. High values are seen for BC over high-albedo surfaces, such as Greenland, the northernmost parts of North America, central Arctic Ocean, and over the Sahara and Arabian deserts. Also, the values are enhanced where much of BC resides above low clouds, e.g. in the norhernmost parts of the Pacific and Atlantic oceans and west of Sahara. In contrast, low values occur where BC is preferentially located below clouds; most notably in the emission region in Fennoscandia where most of the BC resides close to the surface.

**Comment:** *11. Line 377: The longitudinal variation seems interesting, and very policy relevant.*

**Response and change in the manuscript:** Thank you, we agree. The first sentence of this paragraph was modified as follows (line 419 in the marked-up manuscript): "The longitude of the emissions also matters, which should be considered in climate policy."

**Comment:** *12. Section 6.1: Suggest to also compare with ECLIPSE project, Bellouin et al. 2016.*

**Response:** It is actually not easy to make an apple-to-apple comparison with Bellouin el al. (2016), for various reasons (only two emission regions, which naturally do not match with ours; specific radiative forcings for aerosol-radiation and aerosol-cloud interaction lumped

together, etc.). However, the seasonal variations can be mentioned, and also the comparison between European and East Asian emissions.

**Change in the manuscript**: The following excerpts have been added to the first paragraph of Sect. 6.1 (lines 387–389 and 399–406 in the marked-up manuscript): "The findings regarding emission season (larger dirSF for BC emissions in summer than in winter, and vice versa for snowSF except for permanently snow-covered regions) are largely consistent with previous work (Bond et al. 2011; Bellouin et al. 2016). . . . The details sometimes differ. For example, while Fig. 4a suggests that BC dirSF is larger for East Asian than European emissions, Fig. 6 in Bellouin et al. (2016) shows broadly similar specific RF for airborne BC for these two emission regions. This comparison is slightly ambiguous because Bellouin et al. (2016) did not report separately the contributions from aerosol-radiation and cloud-radiation interaction. However, the difference could also be related to the experimental setup. Bellouin et al. (2016) considered 20% reductions from near-present baseline emissions (which makes the baseline substantially more polluted for East Asia), while we added equally strong BC emissions in each region on top of a zero-BC baseline case. Furthermore, based on the results of Bellouin et al. (2016), snowSF is more than twice as large for European emissions as for Asian emissions, while in our results (Fig. 4b) this difference is less obvious."

**Comment:** *13. Section 6.2: Suggest to also compare with ECLIPSE project, Aamaas et al. 2016 and 2017.*

**Response:** A qualitative comparison to European and East Asian metrics represented in Aamaas et al. (2016, 2017) has been added.

**Change in the manuscript**: The following text has been added to Section 6.2 (lines 485–491 in the marked-up manuscript): "The normalized temperature response in Fig. 7c can also be qualitatively compared with Aamaas et al. (2016, 2017), who computed Global Temperature Potential (GTP) and Absolute Regional Temperature Potential metrics for BC emissions in Europe and East Asia based on the RF data from Bellouin et al. (2016). Overall, their results suggest that the global-mean temperature is more sensitive to changes in European than East Asian BC emissions, especially if the enhanced efficacy of BC snowRF is taken into account (Aamaas et al. 2017). In contrast, our results rather show a slightly larger $\Delta T_{\mathrm{glob,norm}}$ for BC emissions in East Asia than in Europe. These differences are consistent with the respective RF differences to Bellouin et al. (2016) discussed above."

**Comment:** *14. Lines 389: This paragraph would be clearer if it included discussion of meteorological adjustments and ERFs. The reason dirRF for BC has less effect on climate*

*is not because it has lower "efficacy", it is because there are adjustments that oppose the dirRF so that the ERF is lower than dirRF (e.g. Stjern et al. 2017). Studies of BC efficacy defined in terms of ERF (E.g. Richardson et al. 2019) show an efficacy of around 1.0 when compared to CO2.*

**Response:** We agree that the relatively small temperature response to BC dirRF is a consequence of rapid adjustments, which act to reduce the BC effective radiative forcing (ERF). Thus, the low efficacy pertains to the case when the efficacy is defined wrt. the instantaneous RF (or the stratosphere-adjusted RF, which is nearly the same for BC) but indeed not necessarily when it is defined wrt. ERF. We have modified this paragraph to make clearer the physical origins of the muted temperature response to dirRF (and also the enhanced temperature response to snowRF).

**Change in the manuscript**: The first paragraph of Section 6.2 has been modified (lines 431–445 in the marked-up manuscript, 409–421 in the revised manuscript). Please see our response to Your comment #2.

**Comment:** *15. Figure 7: This figure is presumably very sensitive to the assumed factor of 3 efficacy for snowRF. What is the uncertainty in this factor of 3? Would the conclusions be qualitatively the same with a lower factor?*

**Response:**
To be precise, this factor of 3 is the ratio of efficacies for snowRF and dirRF. It is difficult to give an uncertainty range for this factor, especially because the optimal value of this factor probably depends on the location of BC emissions. However, all studies that were are aware of suggest that the efficacy of BC snowRF (when defined wrt. stratosphere-adjusted RF, which is almost the same as the instantaneous RF for BC) is well above 1. Hansen and Nazarenko (2004) give an efficacy of roughly two, Hansen et al. (2005) 1.71, and Flanner et al. (2007) 2.1–4.5 depending on the experiment. Furthermore, the efficacy for BC dirRF is below 1 (again, when defined wrt. wrt. the instantaneous or stratosphere-adjusted RF). Most relevantly, Hansen et al. (2005) report values of 0.58–0.78 for GISS ModelE-R, which is the model used by Shindell and Faluvegi (2009) to derive the RCS coefficients that we employ for BC dirRF. Therefore, the value of 3 seems well justified, or perhaps even conservative, if one puts emphasis on the results of Flanner et al. (2007). Yet, on physical grounds it could be speculated that the factor of 3 may be too high for regions with permanent snow cover, such as Antarctica and the interior of Greenland. In those regions the BC-induced change in snow cover is eliminated, which reduces greatly the snow albedo feedback and presumably also the impact on boundary layer stability.

For checking the impact of a smaller efficacy ratio, Fig. 4 below shows the normalized

temperature response and fractional contribution of snowRF to it, for efficacy ratios of 3 (uppermost row, similar to Fig. 7c,d in the manuscript), 2 (middle row), and 1 (lowermost row). For the rather extreme assumption that the efficacy ratio is 1, the normalized temperature response in the Arctic is reduced by about 30% and that in the Antarctica by 60-65% compared to that for our default efficacy factor of 3. Nevertheless, the qualitative features remain largely the same. (Note that the reduction is less in the Arctic because for the Arctic temperature response to local BC RF in the Arctic, values modelled by Flanner (2013) are used).

**Change in the manuscript**: The following text hass been added to Section 6.2 (lines 462–474 in the marked-up manuscript): "One uncertainty factor in our quantitative results is the assumption that the efficacy for BC snowRF is 3 times as large as that for BC dirRF (except for the Arctic temperature response to local BC RF in the Arctic, which was modelled explicitly by Flanner (2013)). Previous studies have indicated that the efficacy of BC snowRF defined wrt. the stratospheric adjusted RF (which is almost the same as the instantaneous RF) is well above 1; Hansen and Nazarenko (2004) give an efficacy of roughly 2, Hansen et al. (2005) 1.71, and Flanner et al. (2007) 2.1–4.5, depending on the experiment. Furthermore, as noted above, due to rapid adjustments, the efficacy of BC dirRF defined wrt. the stratospheric adjusted RF is very likely below 1. In particular, Hansen et al. (2005) report values of 0.58–0.78 for GISS ModelE-R, which Shindell and Faluvegi (2009) employed to derive the RCS coefficients used for BC dirRF here. Therefore the value of 3 for the snowRF-vs-dirRF efficacy ratio seems well justified. Yet it might be an overestimate for regions with permanent snow (such as Antarctica and the interior of Greenland), where BC-induced snow cover changes are absent, and therefore, positive feedbacks due to albedo and lapse-rate changes are probably greatly reduced. If the snowRF-vs-dirRF efficacy ratio were set to 1, which is almost certainly too low, the normalized temperature response in Fig. 7c would be reduced by $\sim$30% for emissions in the Arctic and by 60-65% for emissions in the Antarctica; yet even then the qualitative features seen in Figs. 7c,d would remain largely the same."

**Comment:** *16. Line 518: Suggest to use "meteorological adjustments" or "rapid adjustments" rather than "semi-direct effect" to align with IPCC terminology.*

**Response and change in the manuscript:** This has been reformulated as "...ignoring rapid adjustments and BC indirect effects on clouds" (line 598 in the marked-up manuscript).

[Figure]

Figure 4: (left) Estimated global-mean temperature response normalized by the BC emissions (mK $\left(\text{kg s}^{-1}\right)^{-1}$) and (right) the fractional contribution of snowRF to the temperature response (%) as a function of BC emission region. (a–b): The uppermost row ("3X") represents the case in which the snowRF-vs-dirRF efficacy ratio is set to 3, as in Figs. 7c and 7d of the manuscript. In the middle row (c–d, "2X"), this ratio is set to 2, and in the lowermost row, (e–f, "1X"), it is set to 1. As an exception, for estimating the Arctic temperature to local radiative forcing in the Arctic (north of 60°N), RCS values derived by Flanner (2013) are used.